# Addressing Imbalanced Domain-Incremental Learning through Dual-Balance Collaborative Experts

**Lan Li** [1 2]   **Da-Wei Zhou** [1 2]   **Han-Jia Ye** [1 2]   **De-Chuan Zhan** [1 2]

## Abstract

Domain-Incremental Learning (DIL) focuses on continual learning in non-stationary environments, requiring models to adjust to evolving domains while preserving historical knowledge. DIL faces two critical challenges in the context of imbalanced data: intra-domain class imbalance and cross-domain class distribution shifts. These challenges significantly hinder model performance, as intra-domain imbalance leads to underfitting of few-shot classes, while cross-domain shifts require maintaining well-learned many-shot classes and transferring knowledge to improve few-shot class performance in old domains. To overcome these challenges, we introduce the Dual-Balance Collaborative Experts (DCE) framework. DCE employs a frequency-aware expert group, where each expert is guided by specialized loss functions to learn features for specific frequency groups, effectively addressing intra-domain class imbalance. Subsequently, a dynamic expert selector is learned by synthesizing pseudo-features through balanced Gaussian sampling from historical class statistics. This mechanism navigates the trade-off between preserving many-shot knowledge of previous domains and leveraging new data to improve few-shot class performance in earlier tasks. Extensive experimental results on four benchmark datasets demonstrate DCE's state-of-the-art performance.

## 1. Introduction

In recent years, deep learning has demonstrated powerful representation learning capabilities in various fields, including computer vision and natural language processing (He

[1]School of Artificial Intelligence, Nanjing University, China [2]National Key Laboratory for Novel Software Technology, Nanjing University, China. Correspondence to: Da-Wei Zhou <zhoudw@lamda.nju.edu.cn>.

*Proceedings of the $42^{nd}$ International Conference on Machine Learning*, Vancouver, Canada. PMLR 267, 2025. Copyright 2025 by the author(s).

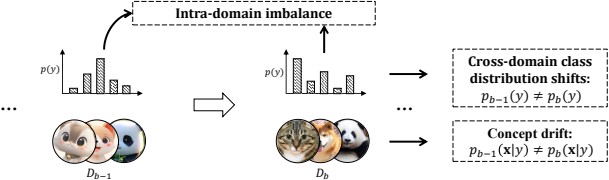

*Figure 1.* A schematic illustration of imbalanced DIL: In addition to concept drift (e.g., image style discrepancies), imbalanced DIL exhibits intra-domain class imbalance (varying sample quantity ratios within individual tasks) and cross-domain class distribution shifts (class distribution differences across domains).

et al., 2016; Deng et al., 2009; Floridi & Chiriatti, 2020). However, the real world is inherently dynamic, with data often arriving in non-stationary streams (Aggarwal, 2018; Ye et al., 2024; Zhuang et al., 2023), requiring models to continuously adapt to changes in data distributions (Gama et al., 2014). A core challenge in this context is catastrophic forgetting, where models forget previously learned knowledge while adapting to new data. To address this, Domain-Incremental Learning (DIL) has emerged as a critical paradigm in continual learning, focusing on model adaptation in the presence of concept drift (Han et al., 2021). Recent studies have made significant progress in mitigating forgetting caused by domain shifts by leveraging pre-trained models (PTMs) (Wang et al., 2022c;b;a). These approaches typically initialize DIL models using PTMs, freeze their parameters to preserve existing knowledge, and introduce lightweight modules (Jia et al., 2022; Chen et al., 2022; Lian et al., 2022) to adapt to new domains.

However, class imbalance, a pervasive issue in real-world data, remains an underexplored challenge in the presence of concept drift (Yang et al., 2022). For example, in autonomous driving systems, traffic sign samples under extreme weather conditions are much rarer than those in normal scenarios. Class imbalance in DIL manifests in two dimensions: intra-domain imbalance and cross-domain class distribution shifts, as shown in Figure 1.

Imbalanced CIL introduces two major challenges for existing PTM-based DIL methods. First, intra-domain class imbalance causes models to overfit many-shot classes during training, leading to underrepresentation and poor general-

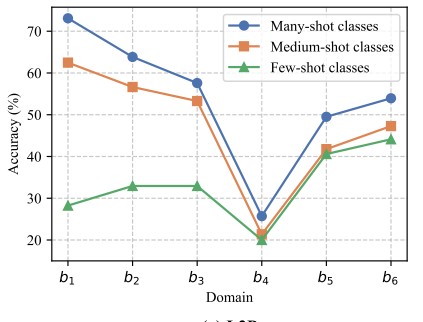 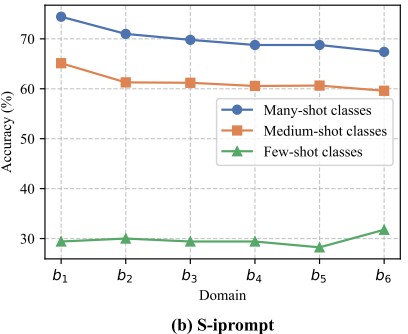 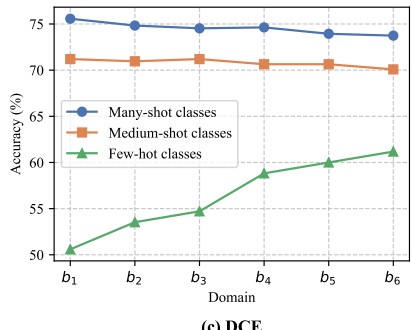

| (a) L2P | (b) S-iprompt | (c) DCE |

*Figure 2.* On the DomainNet dataset, we illustrate the accuracy changes for test samples from the first domain during the training process (denoted by domain $b_1$ to $b_6$). The shared prompt method (L2P) benefits from a shared feature space during updates, resulting in improved performance for few-shot classes, but suffers from catastrophic forgetting of many-shot classes. In contrast, the domain-specific prompt method (S-iprompt) avoids knowledge sharing between tasks, resulting in lower catastrophic forgetting for many-shot classes but no performance improvement for few-shot classes in the new domain. Our DCE reduces forgetting in many-shot classes and improves performance for few-shot classes.

ization on few-shot classes. Second, cross-domain class distribution shifts pose a more complex problem that goes beyond simply mitigating catastrophic forgetting. In this setting, the model must not only retain well-learned knowledge from previous tasks but also effectively transfer new-task information to improve few-shot class performance. However, mechanisms designed to prevent forgetting often inadvertently suppress the generalization potential of few-shot classes by limiting their ability to benefit from new-domain data, as shown in Figure 2. This requires DIL methods to simultaneously preserve the performance of well-learned many-shot classes and enhance the generalization of few-shot classes through knowledge sharing across domains.

To address these challenges, we propose Dual-Balance Collaborative Expert (DCE) with a dual-phase training paradigm. In the first stage, we train multiple frequency-aware expert networks, each specialized in learning representations for a specific frequency group, which helps balance the representations of many-shot and few-shot classes and mitigates underfitting of few-shot classes. In the second stage, we develop a dynamic expert selector trained on synthetic pseudo-features, which are generated through balanced Gaussian sampling based on historical class centroids and covariance matrices. This probabilistic routing mechanism enables context-aware expert collaboration, allowing the model to preserve knowledge of many-shot classes while leveraging patterns learned from new domains to enhance the representation of few-shot classes from previous tasks. It thus achieves a balance between knowledge sharing and resistance to forgetting.

Our contributions are threefold: (1) To the best of our knowledge, we are the first to systematically study the impact of class imbalance in PTM-based DIL, addressing both intra-domain imbalance and cross-domain class distribution shifts. (2) We propose DCE , a framework that employs frequency-

aware experts to resolve intra-domain class imbalance, combined with a dynamic expert selector that balances cross-domain knowledge fusion while mitigating catastrophic forgetting. (3) We establish four benchmarks for imbalanced DIL scenarios, where experimental results demonstrate that DCE achieves state-of-the-art performance. Code is released at `https://github.com/Lain810/DCE`.

## 2. Related Work

**Continual Learning/Incremental Learning** represents a key area in machine learning, with a primary focus on enabling models to assimilate new knowledge from sequential data streams (De Lange et al., 2021; Masana et al., 2023; Wang et al., 2024; Zhao et al., 2021; Zheng et al., 2024; Qi et al., 2025; Ruru et al., 2024). This field is commonly divided into three branches: task-incremental learning (TIL) (De Lange et al., 2021), class-incremental learning (CIL) (Masana et al., 2023; Zhuang et al., 2022; 2024), and domain-incremental learning (DIL) (van de Ven et al., 2022). Both TIL and CIL address challenges associated with the introduction of novel classes, requiring systems to learn new concepts without erasing previously acquired knowledge. In contrast, DIL handles scenarios where the label set remains fixed, but the incoming data originates from new or varying domains (Wang et al., 2022a; Shi & Wang, 2023).

**Domain-Incremental Learning with Pre-Trained Models:** The surge in pre-training methodologies has revolutionized DIL by offering robust initializations that enhance feature transferability and generalization (Wang et al., 2022b;c;a; Kim et al., 2023; Smith et al., 2023). Recent DIL approaches that leverage pre-trained models (PTMs) primarily focus on visual prompt tuning (Jia et al., 2022), where the pre-trained feature extractor is frozen, and a prompt pool is learned as external knowledge. These prompts are selectively applied to encode instance-specific or task-specific

information. For instance, L2P (Wang et al., 2022c) employs a query-key matching mechanism for selecting appropriate prompts, while DualPrompt (Wang et al., 2022b) refines this strategy by jointly learning prompts specific to tasks and instances. In another approach, S-Prompts (Wang et al., 2022a) emphasizes domain-specific prompts, utilizing KNN-based retrieval for prompt selection. To streamline the prompt retrieval process, CODA-Prompt (Smith et al., 2023) introduces an attention-driven weighted combination to replace conventional selection mechanisms. Beyond prompt tuning, several alternative strategies have been proposed. These include directly constructing classifiers based on pre-trained features (Zhou et al., 2024a; McDonnell et al., 2023), merging models to integrate knowledge across tasks (Zhou et al., 2024b), and designing Mixture-of-Experts architectures (Yu et al., 2024) that dynamically select specialized experts. Among them, Yu et al. (2024) introduced a Mixture-of-Experts Adapter framework for vision-language models. In contrast, our method adopts a different expert structure and does not rely on a language model. Furthermore, none of these approaches explicitly address the class imbalance issue inherent in DIL, which our method is designed to handle.

**Class-Imbalanced learning**: A lot of research has explored class-imbalanced learning (Liu et al., 2019; Zhang et al., 2023b). Proposed approaches typically include strategies like re-balancing the data through over-sampling few-shot classes or under-sampling many-shot classes (Chawla et al., 2002; He et al., 2008; Mingjian et al., 2022), adjusting or re-weighting loss functions (Cui et al., 2019; Cao et al., 2019; Menon et al., 2020; Ren et al., 2020; Ye et al., 2020), and leveraging various learning paradigms. These paradigms encompass transfer learning (Liu et al., 2019), metric learning (Zhang et al., 2017), meta-learning (Shu et al., 2019), two-stage training (Kang et al., 2020), ensemble learning (Wang et al., 2020; Zhang et al., 2021), self-supervised learning (Yang & Xu, 2020; Li et al., 2022) and knowledge distillation (Li et al., 2024). Compared with these works, we focus on the class imbalance learning problem in DIL, where new challenges need to be addressed.

# 3. Imbalanced Domain-Incremental Learning

This section provides an overview of domain incremental learning, examines the challenges posed by class imbalance, and critiques existing incremental learning methods in the context of imbalanced data.

## 3.1. Background

Domain-incremental learning (DIL) (Zhou et al., 2023a) is a subfield of incremental learning where a model encounters a sequence of tasks, each associated with a distinct domain. Formally, in DIL, the model encounters a sequence of tasks

$\{\mathcal{D}^1, \mathcal{D}^2, \cdots, \mathcal{D}^B\}$, where each $\mathcal{D}^b = \{\mathcal{X}_b, \mathcal{Y}_b\}$ represents the $b$-th training task. Here, $\mathcal{X}_b = \{\mathbf{x}_i\}_{i=1}^{n_b}$ denotes the input instances, and $\mathcal{Y}_b = \{y_i\}_{i=1}^{n_b}$ denotes the corresponding labels, with each input instance $\mathbf{x}_i \in \mathbb{R}^D$ belonging to a class $y_i \in \mathcal{Y}$. Let $n_b^c$ represent the number of samples belonging to class $c$ in domain $b$, and $p_b(y)$ represent the class distribution in domain $b$, where $c \in \mathcal{Y}$. Unlike traditional incremental learning scenarios where the label space may expand, DIL maintains a fixed label space $\mathcal{Y}$ throughout the learning process. However, the input distribution varies across domains introducing significant domain shifts that the model must adapt to without forgetting previously acquired knowledge. By decomposing the joint distribution as $p_b(\mathbf{x}, y) = p_b(y)p_b(\mathbf{x}|y)$, we see that domain shifts include both concept drift (from $p_b(\mathbf{x}|y)$) and class distribution shifts (from $p_b(y)$). Current research often focuses on the former while neglecting the latter. In real-world scenarios, the difficulty of data collection for each class across different domains varies, resulting in discrepancies in their quantities. From the perspective of each individual domain, $p_b(y)$ may follow an imbalanced distribution, which we term intra-domain imbalance. From an inter-domain perspective, the variation of $p_b(y)$ across domains manifests as cross-domain class distribution shift. For simplicity, we define this setting as imbalanced DIL and conduct our study within this framework.

## 3.2. Dilemmas of Current Methods

In this paper, we focus on PTM-based methods in the exemplar-free DIL setting. Generally, the model can be divided into two main components: the feature encoder and the linear classifier. The feature encoder $\theta(x) : \mathbb{R}^D \to \mathbb{R}^d$ is initialized with the parameters $\theta_0$ of a pre-trained Vision Transformer (ViT) (Dosovitskiy et al., 2020), as assumed in (Wang et al., 2022b;c;a; Smith et al., 2023). Additionally, the feature encoder contains some adjustable parameters $\theta_1$, such as prompts, adapters, or feature transformations, to help the model adapt to different domains.

In DIL, when encountering a new domain, the common approach to reduce catastrophic forgetting of previous domains while capturing discriminative features within the new domain is to freeze the pre-trained weights and fine-tune the adjustable parameters $\theta_1$ on the new domain's data to encode domain-specific knowledge. Formally, the optimization objective is:

$$\min_{\theta_1, h} \sum_{(\mathbf{x}, y) \in \mathcal{D}^b} \ell\left(h\left(\theta(\mathbf{x})\right), y\right) + \mathcal{L}_{\theta_1}, \tag{1}$$

where $\mathcal{D}^b$ is the dataset for the $b$-th domain, $h$ is the classifier, $\ell$ is the loss function, and $\mathcal{L}_{\theta_1}$ is a regularization term for the adjustable parameters $\theta_1$. In DIL with PTMs, existing approaches can be categorized into two paradigms based on prompt learning strategies (Wang et al., 2022a):

**Shared prompt paradigm.** Taking L2P (Wang et al., 2022c) as an example, it maintains a learnable prompt pool as shared knowledge across domains. Specifically, domain-specific knowledge is encoded in parameterized prompt sets $\theta_1$, while the prompt selection mechanism is guided by $\mathcal{L}_{\theta_1}$. This paradigm adapts feature representations through retrieved prompts and progressively extends the classifier dimension with emerging tasks. The final prediction is adjusted using a modulo operation over the label space size $|\mathcal{Y}|$. However, continuously updating knowledge within a shared feature space inherently causes entanglement between old and new knowledge subspaces, leading to degraded class separability.

**Domain-specific prompt paradigm.** Methods such as S-iPrompt (Wang et al., 2022a) learn independent sets of parameters $\theta_1^b$ for each domain $b$. They establish a cluster-based domain identification mechanism: extracting domain features via PTM, clustering them into domain prototypes, and performing hard assignment during inference by measuring similarity between test samples and prototypes. While this architecture mitigates knowledge interference through parameter isolation, its strictly segregated design faces critical limitations — the absence of explicit cross-domain knowledge sharing mechanisms may result in suboptimal generalization capability.

**Discussions:** While both paradigms suffer from class imbalance, their underlying mechanisms exhibit distinct characteristics. As shown in Figure 2, intra-domain class imbalance affects both paradigms similarly. For few-shot classes in the training domain $b_1$, models tend to misclassify samples as many-shot classes due to underfitting, resulting in significant accuracy disparities between frequency groups.

But cross-domain class distribution shift reveals a significant difference between shared prompt and domain-specific prompt methods. Shared prompts maintain a common prompt pool and feature space across tasks, which facilitates knowledge transfer and improves few-shot class performance on earlier domains after training new tasks. However, this shared space also increases conflicts between old and new tasks, resulting in more forgetting for many-shot classes, as shown in Figure 2(a). In contrast, domain-specific prompts isolate prompts and feature spaces for each task, effectively reducing forgetting for many-shot classes when the test sample's task is correctly identified. Nevertheless, this isolation limits the transfer of knowledge to few-shot classes from new tasks, thereby restricting their generalization, as illustrated in Figure 2(b). In short, although shared prompts enhance few-shot transfer, they carry a higher risk of forgetting many-shot classes, while domain-specific prompts reduce forgetting for many-shot classes at the expense of weaker few-shot generalization.

Thus, we believe that in imbalanced DIL, the model should possess two key capabilities to address the challenges posed by intra-domain imbalance and cross-domain distribution shifts. **(1) For intra-domain imbalance, the model should balance learning across different classes during training, reducing the bias towards many-shot classes. (2) For cross-domain class distribution shifts, the model must strike a balance between preserving prior knowledge and acquiring new information.** Specifically, the model should retain the discriminative ability of well-learned many-shot classes to prevent forgetting, while allowing few-shot classes to benefit from new task data and improve generalization without being constrained by outdated knowledge.

## 4. Methodology

The following sections detail our Dual-Balance Collaborative Experts (DCE) framework, which includes two training stages per domain, designed to address the limitations of both shared and domain-specific prompt paradigms. In the first stage, we construct frequency-aware expert networks to isolate task-specific knowledge, with each expert specializing in different class frequency groups. This design mitigates intra-domain imbalance and alleviates forgetting due to shared feature spaces. In the second stage, we develop a dynamic expert selector that assigns soft weights to the contributions of each expert. The selector is trained on balanced pseudo-features sampled from historical class statistics. This mechanism encourages knowledge sharing across tasks, enabling underrepresented classes from previous domains to benefit from new task data, thereby improving generalization without compromising stability.

### 4.1. Frequency-Aware Experts

In imbalanced DIL, intra-domain class imbalance leads to excessive bias toward many-shot classes while insufficient representation learning for few-shot classes. To address this, we propose a multi-expert collaborative architecture that constructs specialized expert networks for different frequency classes through differentiated training strategies. As shown in Figure 3, we deploy three parallel expert modules downstream of the PTM, denoted as $e_b$, $e_{b+1}$, and $e_{b+2}$, targeting many-shot-biased, balanced, and few-shot-biased learning respectively. Each expert consists of an MLP and a dedicated classifier.

For the first expert $e_b$, we employ the standard softmax cross-entropy loss:

$$\ell_{\text{CE}} = - \log \frac{\exp(v_y^1)}{\sum_{j \in |\mathcal{Y}|} \exp(v_j^1)}, \tag{2}$$

where $v^1 = e_b(\theta(\mathbf{x}))$. This loss assumes distributional alignment between training and test data, which is violated in class-imbalanced settings. As a result, it tends to favor frequent classes.

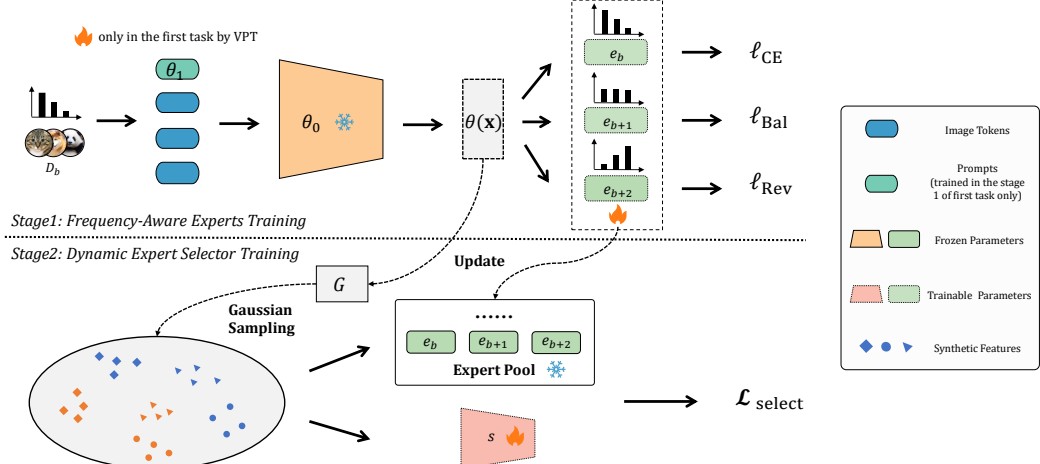

*Figure 3.* The overall pipeline of our proposed DCE , consists of a two-stage training process: frequency-aware experts training and dynamic expert selector training. In the first stage, each expert is trained independently with its own loss function, where the visual prompt is learned in the first task and frozen thereafter. In the second stage, we compute class-wise means and covariances using the frozen feature extractor, then perform Gaussian sampling to construct a synthetic feature set, which is used to train the expert selector.

To address this, the second expert $e_{b+1}$ uses the balanced softmax loss (Ren et al., 2020), which corrects the bias by incorporating class frequency priors from the imbalanced training distribution $p_b(y) = \{p_b^i\}_{i=1}^{|\mathcal{Y}|}$:

$$\ell_{\text{Bal}} = -\log \frac{\exp(v_y^2 + \log p_b^y)}{\sum_{j \in |\mathcal{Y}|} \exp(v_j^2 + \log p_b^j)}. \quad (3)$$

Different from $\ell_{\text{CE}}$ that prioritizes many-shot classes, $\ell_{\text{Bal}}$ encourages balanced predictions by compensating for skewed label frequencies. we further propose an inverse distribution loss $\ell_{\text{Rev}}$ for the third expert $e_{b+2}$. This loss inverts the training distribution to emphasize few-shot classes, using a normalized inverse frequency distribution $\hat{p}_b(y) = \{\hat{p}_b^i\}_{i=1}^{|\mathcal{Y}|}, \hat{p}_b^i = \frac{1}{p_b^i} / \sum_j^{\mathcal{Y}} (\frac{1}{p_b^j})$. Therefore we have:

$$\begin{aligned} \ell_{\text{Rev}} &= -\log \frac{\exp(v_y^3 + \log p_b^y - \log \hat{p}_b^y)}{\sum_{j \in |\mathcal{Y}|} \exp(v_j^3 + \log p_b^j - \log \hat{p}_b^j)} \\ &= -\log \frac{\exp(v_y^3 + 2\log p_b^y)}{\sum_{j \in |\mathcal{Y}|} \exp(v_j^3 + 2\log p_b^j)}. \end{aligned} \quad (4)$$

The proof of Equation (4) is provided in Appendix A. This loss induces a stronger focus on few-shot classes, providing complementary behavior to $e_b$.

During training, each expert is optimized independently using its designated loss. The overall training objective for expert modules is the sum of all three:

$$\ell_{\text{exp}} = \ell_{\text{CE}} + \ell_{\text{Bal}} + \ell_{\text{Rev}}. \quad (5)$$

### 4.2. Dynamic Expert Selector

After training task-specific experts, the central challenge lies in selecting an optimal combination of experts for test

samples with unknown domain affiliation. While these experts alleviate catastrophic forgetting by isolating conflicting knowledge subspaces, naive expert selection remains problematic. For example, domain-specific approaches such as S-iPrompt (Wang et al., 2022a) rely on a hard assignment strategy, matching each test sample to a single domain expert based on feature similarity to pre-computed domain prototypes, limiting flexibility and adaptability as the expert set grows. This static strategy is insufficient in the presence of cross-domain label distribution shifts, where newly trained experts may offer stronger representations for few-shot classes from previous domains. To address this, we propose a dynamic expert selection mechanism that evolves with the expanding expert pool, enabling adaptive fusion of cross-domain knowledge.

We use a MLP $s(\cdot) : \mathbb{R}^d \rightarrow \mathbb{R}^{3b}$ as the dynamic expert selector at task $b$. Given a sample $\mathbf{x}$, the selector takes the feature $\theta(\mathbf{x})$ as input and outputs a weight vector $\mathbf{w} = [w_i]_{i=1}^{3b}$, where each $w_i$ denotes the importance of expert $e_i$. Here, $3b$ corresponds to the total number of experts accumulated up to task $b$.

To train the expert selector, we first construct a task-relevant and stable feature space using the frozen parameters of the pre-trained model. Specifically, during the training of the first task's experts, we employ Visual Prompt Tuning (VPT) (Jia et al., 2022) on the encoder $\theta$, where the backbone parameters $\theta_0$ are kept frozen and only the prompt parameters $\theta_1$ are optimized to obtain a stable and DIL-aware feature space. In subsequent tasks, both $\theta_0$ and $\theta_1$ are frozen to maintain feature consistency across tasks.

Following the observation from Zhang et al. (2023a) that PTMs typically yield unimodal feature distributions per

class, we model each class as a Gaussian distribution. During DIL, we continually collect and update feature statistics across domains. After training the experts for domain $b$, we compute and store class-wise feature statistics $G_b = \{(\mu_b^c, \Sigma_b^c)\}$, where $\mu_b^c$ and $\Sigma_b^c$ are the empirical mean and covariance of class $c$. These are then merged into a global statistical repository $G = \bigcup_{i=1}^{b} G_i$, which incrementally captures the evolving cross-domain feature landscape.

During the optimization of the expert selector, cross-domain knowledge transfer is facilitated by sampling features from Gaussian distributions. For each domain-class pair $(b, c)$ in $G$, Gaussian sampling is performed to generate $K$ synthetic features, forming the synthetic dataset:

$$\hat{D} = \bigcup_{i=1}^{b} \bigcup_{c=1}^{|\mathcal{Y}|} \{(\tilde{\mathbf{x}}, c) \sim \mathcal{N}(\mu_i^c, \Sigma_i^c)\}_{k=1}^{K}. \qquad (6)$$

Importantly, the value of $K$ is kept uniform across all domain-class pairs, ensuring balanced knowledge transfer across imbalanced classes and domains. The expert selector $s$ is then updated using:

$$\mathcal{L}_{\text{Select}} = \frac{1}{|\hat{D}|} \sum_{(\tilde{\mathbf{x}}, y) \in \hat{D}} \ell_{\text{CE}} \left( \sum_{i=1}^{3b} s(\tilde{\mathbf{x}})_i \cdot e_i(\tilde{\mathbf{x}}), \ y \right). \quad (7)$$

However, estimating the covariance matrices in imbalanced DIL poses practical challenges. For classes with few samples, the covariance estimation becomes unreliable. To address this, we adopt Oracle Approximating Shrinkage (OAS) (Chen et al., 2010), which introduces a shrinkage mechanism to produce more stable estimates. Furthermore, with the covariance matrix having dimensions of $d \times d$, storing a separate covariance matrix for each class across different domains would require substantial storage space. To reduce storage costs, we average the class-specific covariances within each domain, resulting in a single domain-level covariance matrix. More detail can be found in Appendix B.

### 4.3. Summary of DCE

The proposed DCE is illustrated in Figure 3, and summarized in Algorithm 1. Each task involves two stages: frequency-aware expert training and dynamic expert selector training. In the first stage, we independently train each expert using its corresponding loss function. The prompt parameters $\theta_1$ are optimized via VPT during the first task and kept frozen for all subsequent tasks to ensure consistency. In the second stage, we compute and store the class-wise mean and covariance of the features extracted by the frozen pre-trained model. These statistics are used to synthesize representative features via multivariate Gaussian sampling. The resulting synthetic feature set $\hat{D}$ is then used to train the expert selector through the loss $\mathcal{L}_{\text{Select}}$. During inference,

---

**Algorithm 1** Incremental Training of DCE

**Input:** Incremental datasets: $\{\mathcal{D}^1, \mathcal{D}^2, \cdots, \mathcal{D}^B\}$, Pre-trained embedding: $\theta_0(x)$
**Output:** Incrementally trained model
**for** $b = 1$ **to** $B$ **do**
    Get the incremental training set $\mathcal{D}^b$
    **if** $b = 1$ **then**
        Jointly train prompt $\theta_1$ and experts $e_1, e_2, e_3$ by $\ell_{\text{exp}}$ in Equation (5).
    **else**
        Train experts $e_b, e_{b+1}, e_{b+2}$ by $\ell_{\text{exp}}$ in Equation (5).
    **end if**
    Compute per-class statistics $G_b$ of $\mathcal{D}^b$.
    Update global statistics: $G \leftarrow G \cup G_b$.
    Gaussian sampling on $G$ to construct $\hat{D}$ via Equation (6).
    Train selector $s$ on $\hat{D}$ using $\mathcal{L}_{\text{select}}$ in Equation (7).
**end for**

---

for an input $\mathbf{x}$, the extracted feature $\theta(\mathbf{x})$ is passed to all experts. Their outputs are then weighted and fused by the expert selector $s(\theta(\mathbf{x}))$ to produce the final prediction.

## 5. Experiment

In this section, we conduct experiments on benchmark datasets to compare to existing state-of-the-art methods, and provide a detailed analysis of the experimental results.

### 5.1. Experimental Setup

**Dataset:** Following benchmark settings in PTM-based DIL (Wang et al., 2022b;c; Smith et al., 2023), we evaluate four widely-used DIL benchmark datasets: **Office-Home**(Venkateswara et al., 2017) (4 domains), **Domain-Net**(Peng et al., 2019) (6 domains), **CORe50**(Lomonaco & Maltoni, 2017) (11 domains), and **CDDB-Hard**(Li et al., 2023) (5 domains). It is worth noting that the Office-Home and DomainNet datasets are inherently highly imbalanced. Following Yang et al. (2022), we retain the original imbalanced training distributions for Office-Home and Domain-Net, while constructing balanced test sets by equalizing the number of samples per class. For these two datasets, we categorize classes in each task into many-shot, medium-shot, and few-shot groups using thresholds of 20 and 60 for Office-Home, and 20 and 100 for DomainNet. For CORe50, we follow Cui et al. (2019) and apply distinct class imbalance ratios per domain to create imbalanced training sets. Similarly, in CDDB-Hard, we systematically adjust the positive-to-negative sample ratios to induce per-domain class imbalance. Note that the test set of CORe50 consists of three outdoor sessions that are drawn from a distribution different from the per-task training domains. Additionally,

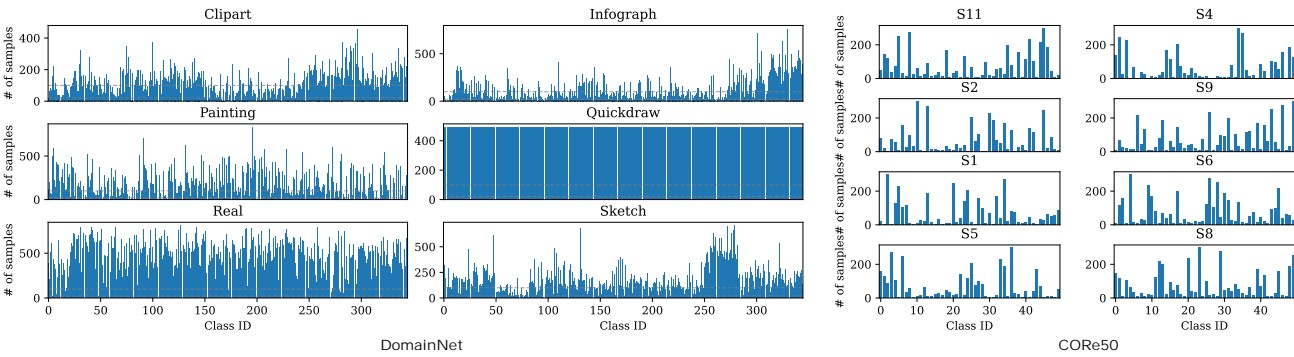

*Figure 4.* Class distribution of training samples in DomainNet and CORe50. DomainNet is inherently imbalanced; hence, we construct a class-balanced test set for evaluation. Dashed lines denote the thresholds for many-shot, medium-shot, and few-shot class divisions. For CORe50, imbalanced training sets are manually created.

*Table 1.* Average and last performance of different methods among five task orders. The best performance is shown in bold. All methods are implemented with ViT-B/16 IN1K. Methods with † indicate implemented with exemplars (10 per class).

| Method | Office-Home | | | | | DomainNet | | | | |
|---|---|---|---|---|---|---|---|---|---|---|
| | $\bar{\mathcal{A}}$ | $\mathcal{A}_B$ | $\mathcal{A}_{many}$ | $\mathcal{A}_{med}$ | $\mathcal{A}_{few}$ | $\bar{\mathcal{A}}$ | $\mathcal{A}_B$ | $\mathcal{A}_{many}$ | $\mathcal{A}_{med}$ | $\mathcal{A}_{few}$ |
| iCaRL† (Rebuffi et al., 2017) | 83.2±3.2 | 83.5±0.7 | **90.5±0.9** | 82.8±0.9 | 66.6±3.1 | 58.4±5.3 | 55.9±1.7 | 58.0±1.3 | 48.3±1.1 | 36.5±3.5 |
| MEMO† (Zhou et al., 2023b) | 78.7±3.7 | 79.1±1.2 | 86.4±1.3 | 78.1±1.3 | 60.5±4.6 | 57.8±6.7 | 56.4±1.4 | 58.3±1.4 | 50.1±1.6 | 38.2±2.3 |
| SimpleCIL (Zhou et al., 2024a) | 75.2±4.9 | 76.2±0.0 | 81.6±0.0 | 74.7±0.0 | 73.7±0.0 | 41.1±6.8 | 40.6±0.0 | 41.5±0.0 | 40.8±0.0 | 30.8±0.0 |
| RanPAC (McDonnell et al., 2023) | 83.4±3.8 | 83.3±0.3 | 89.1±0.5 | 82.1±0.7 | 72.5±3.6 | 57.8±5.5 | 56.1±0.6 | 58.3±0.6 | 52.5±0.4 | 40.0±0.6 |
| L2P (Wang et al., 2022c) | 78.7±4.2 | 80.5±0.6 | 86.1±0.9 | 79.0±1.1 | 73.7±3.9 | 48.5±6.8 | 45.2±2.3 | 46.7±2.3 | 42.0±1.5 | 37.3±0.8 |
| DualPrompt (Wang et al., 2022b) | 77.2±3.1 | 79.1±0.6 | 84.3±0.5 | 77.3±0.7 | 70.6±2.2 | 52.3±9.7 | 50.9±4.3 | 52.5±5.0 | 44.7±3.5 | 37.5±2.4 |
| CODA-Prompt (Smith et al., 2023) | 82.4±3.8 | 83.3±0.3 | 88.7±0.2 | 81.9±0.6 | 73.2±2.0 | 47.6±6.1 | 45.1±1.2 | 46.6±1.3 | 42.2±1.4 | 38.2±1.4 |
| S-iPrompt (Wang et al., 2022a) | 81.4±3.3 | 80.8±0.2 | 88.4±1.1 | 79.1±0.2 | 66.0±0.8 | 59.0±6.8 | 57.9±0.2 | 61.3±0.2 | 47.3±0.7 | 31.5±0.3 |
| DCE | **84.6±3.0** | **84.4±0.2** | 88.7±0.5 | **83.2±0.3** | **79.4±2.5** | **64.3±6.0** | **63.5±0.5** | **65.2±0.6** | **58.6±0.5** | **50.8±0.4** |

*Table 2.* Average and last performance of different methods among five task orders. The best performance is shown in bold. All methods are implemented with ViT-B/16 IN1K. Methods with † indicate implemented with exemplars (10 per class).

| Method | CORe50 | | CDDB-Hard | |
|---|---|---|---|---|
| | $\bar{\mathcal{A}}$ | $\mathcal{A}_B$ | $\bar{\mathcal{A}}$ | $\mathcal{A}_B$ |
| iCaRL† | 71.0±3.7 | 76.0±2.7 | 57.5±9.9 | 54.4±9.9 |
| MEMO† | 66.0±2.7 | 68.2±1.7 | 66.0±2.7 | 68.2±1.7 |
| SimpleCIL | 62.5±1.6 | 67.2±0.0 | 65.5±3.2 | 64.1±1.7 |
| RanPAC | 76.7±1.3 | 78.4±1.7 | 61.7±4.3 | 62.5±2.8 |
| L2P | 72.3±1.3 | 81.7±0.5 | 67.3±5.3 | 65.0±6.5 |
| DualPrompt | 71.0±4.5 | 77.7±1.0 | 66.9±5.8 | 65.6±2.6 |
| CODA-Prompt | 72.8±1.2 | 81.4±1.1 | 67.9±6.5 | 66.6±2.8 |
| S-iPrompt | 62.7±2.2 | 65.8±0.9 | 64.2±5.8 | 63.4±2.8 |
| DCE | **80.1±0.7** | **84.8±0.3** | **74.6±6.5** | **71.8±4.2** |

each task in CDDB-Hard is a binary classification problem. As a result, we do not perform many/medium/few-shot categorization on CORe50 and CDDB-Hard. We evaluate each setting using five randomized domain orders to ensure robustness, with detailed configurations provided in Appendix F. The training distributions of DomainNet and CORe50 are illustrated in Figure 4.

**Comparison methods:** We compare the proposed DCE

against the state-of-the-art CIL/DIL methods. (1) exemplar-based methods, including **iCaRL** (Rebuffi et al., 2017) and **MEMO** (Zhou et al., 2023b), and (2) state-of-the-art exemplar-free PTM-based domain-incremental learning methods, such as **SimpleCIL** (Zhou et al., 2024a), **Ran-PAC** (McDonnell et al., 2023), **L2P** (Wang et al., 2022c), **DualPrompt** (Wang et al., 2022b), **CODA-Prompt** (Smith et al., 2023), and **S-iPrompt** (Wang et al., 2022a). To ensure a fair comparison, all methods are implemented using the same pre-trained backbone network.

**Implementation details:** We deploy the experiments using PyTorch (Paszke et al., 2019) and Pilot (Sun et al., 2025) on NVIDIA 4090. Following (Wang et al., 2022c; Zhou et al., 2024c), we consider two typical pre-trained weights, i.e., ViT-B/16-IN21K and ViT-B/16-IN1K. Both are pre-trained with ImageNet21K (Russakovsky et al., 2015), while the latter is further finetuned on ImageNet1K. We optimize DCE using SGD optimizer with a batch size of 128 for 20 or 30 epochs. The learning rate is set to 0.001. We select 10 exemplars per class for exemplar-based methods using herding (Welling, 2009) algorithm. More detail can be found in Appendix F.

**Performance Measure:** Following (Wang et al., 2022c;a), we denote the accuracy across all seen domains after learn-

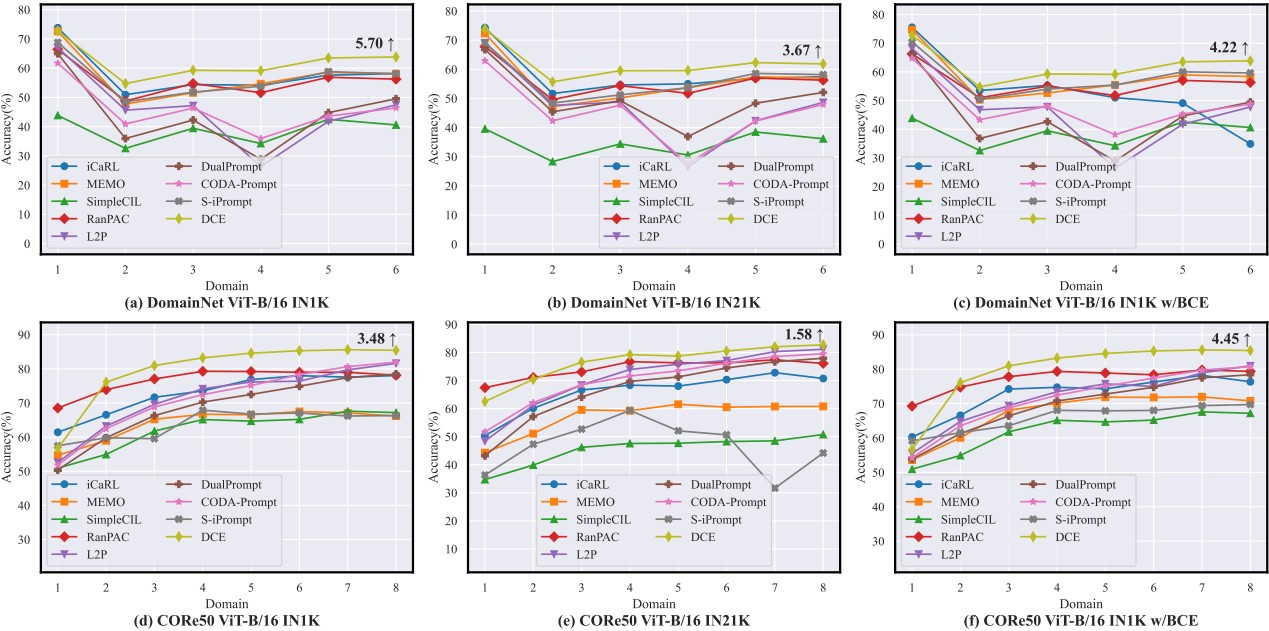

*Figure 5.* Incremental performance of different methods with the same pre-trained model. We report the performance gap after the last incremental stage between DCE and the runner-up method at the end of the line.

ing the $b$-th task as $\mathcal{A}_b$. For comparison, we primarily consider $\mathcal{A}_B$ (the accuracy after completing the final stage) and $\bar{\mathcal{A}} = \frac{1}{B}\sum_{b=1}^{B}\mathcal{A}_b$ (the average accuracy across all incremental stages). Additionally, we evaluate the model's performance across categories with different frequencies: $\mathcal{A}_{\mathrm{many}}$, $\mathcal{A}_{\mathrm{med}}$, and $\mathcal{A}_{\mathrm{few}}$ represent the accuracy on many-shot, medium-shot, and few-shot classes, respectively, at the end of training.

### 5.2. Main Results

We present comprehensive performance statistics (mean ± std) across five task sequences in Table 1 and Table 2, covering four benchmark datasets. From these results, we observe that DCE consistently outperforms comparison methods. Compared to the second-best method, DCE maintains a stable lead on both $\mathcal{A}_B$ and $\bar{\mathcal{A}}$, demonstrating its robustness across different evaluation metrics. In Table 1, we also report the final model's accuracy for many-shot, medium-shot, and few-shot classes separately. Benefiting from the dual-balance approach we adopt, our method shows a significant performance improvement on few-shot classes.

In addition, Figure 5(a) and (d) illustrate performance progression across tasks on DomainNet and CORe50, while Figure 5(b) and (e) show training dynamics using ViT-B/16 pre-trained on ImageNet-21K (IN21K). Our approach demonstrates consistently stable and superior performance, validating its robustness across tasks and compatibility with different pre-trained backbones.

To ensure a fair comparison, we re-implement baseline meth-

ods with the balanced softmax loss $\ell_{\mathrm{Bal}}$. Since these methods are not originally designed for this objective, some of them suffer performance degradation. Nonetheless, as shown in Figure 5(c) and (f), DCE maintains a clear advantage and continues to outperform the baselines under this challenging setting.

### 5.3. Further Analysis

**Class Performance Drift**: In DIL, the conventional forgetting measure is less applicable because class accuracy can increase over time, especially for few-shot classes. To address this, we introduce a new metric, Class Performance Drift (CPD), which quantifies the accuracy change of each class during training. Specifically, for a class $c$ in domain $b$, let $a_b^c$ denote its accuracy immediately after training on domain $b$, and $a_B^c$ denote its accuracy after completing training on the final domain $B$. The CPD is then defined as $\mathrm{CPD}_b^c = a_B^c - a_b^c$, which reflects the performance drift of class $c$ from domain $b$. A positive CPD indicates performance degradation, while a negative value signifies performance improvement.

In Figure 6, we report the mean and variance of CPD on DomainNet across five runs, covering many-shot, medium-shot, few-shot, and all classes. The results are consistent with our earlier analysis. Shared prompt methods (e.g., L2P, DualPrompt, CODA-Prompt) show higher CPD for many-shot and medium-shot classes, indicating more severe forgetting, but lower CPD for few-shot classes, suggesting performance gains from new domain data. In comparison,

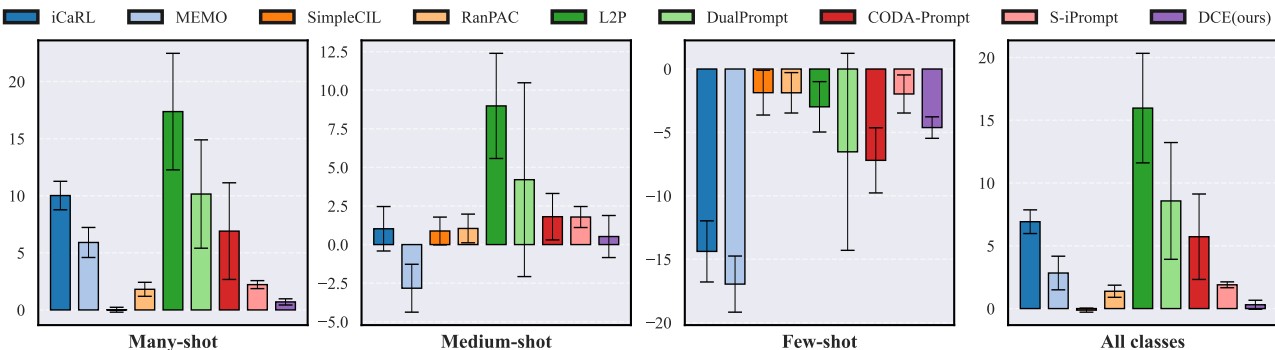

*Figure 6.* Class Performance Drift (lower is better) of different methods on DomainNet dataset among five task orders. DCE demonstrates a balance between reducing forgetting of many-shot classes and improving the performance of few-shot classes.

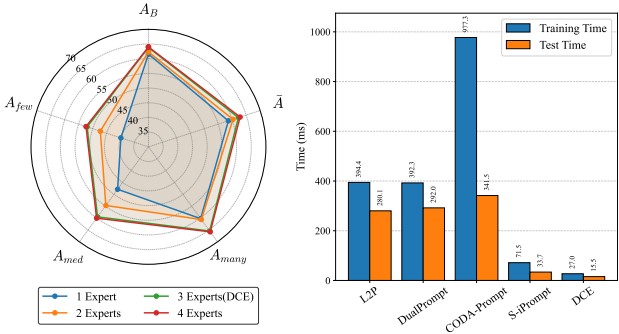

*Figure 7.* Left: On the DomainNet dataset, the performance of the model with different numbers of experts per task. Right: The per-batch training and inference time of different methods on the DomainNet dataset.

domain-specific prompt methods (e.g., S-iPrompt) achieve lower CPD for many-shot and medium-shot classes, indicating better knowledge retention, but exhibit higher CPD for few-shot classes, implying less benefit from new tasks. DCE provides a more balanced outcome, maintaining moderate CPD across all class types. It reduces forgetting in common classes while still improving the performance of rare ones. In terms of overall CPD, our method ranks second only to SimpleCIL, which does not involve training and relies solely on class prototypes. This demonstrates the effectiveness of our dual-balance strategy in balancing knowledge retention and adaptation.

**Effectiveness of Multiple Experts**: The left part of Figure 7 presents a systematic evaluation of the impact of using different numbers of experts. We compare the model's performance under four configurations: We compare the model's performance under four configurations: (1) a single expert trained with $\ell_{CE}$; (2) two experts trained with $\ell_{CE}$ and $\ell_{Bal}$; (3) three experts with an additional $\ell_{Rev}$ (i.e., our proposed DCE configuration), and (4) four experts, where an extra loss $\ell_4 = -\log \frac{\exp(v_y^4 + 3\log p_b^y)}{\sum_{j \in |\mathcal{Y}|} \exp(v_j^4 + 3\log p_b^j)}$ is introduced in addition to the previous three losses. The results show that as the number of experts increases, overall model performance

improves, particularly on medium-shot and few-shot classes. This confirms the effectiveness of our multi-expert strategy. However, the performance gain becomes marginal when adding more than three experts. Therefore, our method adopts a three-expert configuration to strike a balance between effectiveness and efficiency. More discussion can be found in Appendix E.

**Computational Efficiency:** The right part of Figure 7 presents the training and inference costs of different methods. All experiments are conducted on an RTX 3090 GPU, and we report the average per-batch time under the same batch size. For DCE, the average per-batch training time after the first task is computed by summing the durations of both the first and second training stages. Since DCE updates only expert parameters after the first task, the gradient does not propagate through the feature encoder, which eliminates the need to construct a computation graph over it. Additionally, unlike methods such as L2P and DualPrompt that require two forward passes through the encoder for both training and inference, DCE requires only a single forward pass. As a result, our method significantly reduces computational cost in both phases.

## 6. Conclusion

In this work, we present DCE, a novel framework for Domain-Incremental Learning under class-imbalanced conditions. DCE addresses two fundamental challenges: intra-domain class imbalance and cross-domain class distribution shifts, both of which frequently occur in real-world scenarios. By introducing frequency-aware collaborative experts trained with specialized objectives, our method effectively balances the representation of many-shot and few-shot classes during domain-specific learning. Furthermore, the dynamic expert selector, guided by pseudo-features synthesized from historical statistics, enables adaptive knowledge transfer across domains while mitigating forgetting. Extensive experiments on four benchmark datasets validate the effectiveness and efficiency of DCE.

# Acknowledgements

This work is partially supported by National Science and Technology Major Project (2022ZD0114805), NSFC (62376118, 62250069), Fundamental Research Funds for the Central Universities (14380021, 2024300373), the AI & AI for Science Project of Nanjing University, Postgraduate Research & Practice Innovation Program of Jiangsu Province, Collaborative Innovation Center of Novel Software Technology and Industrialization.

# Impact Statement

This paper presents work whose goal is to advance the field of Machine Learning. There are many potential societal consequences of our work, none which we feel must be specifically highlighted here.

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

# A. Proof of Equation (4)

Let the target class probability distribution be $\hat{p}(y) = \{\hat{p}^i\}_{i=1}^{|\mathcal{Y}|}$, and the source class probability distribution be $p(y) = \{p_b^i\}_{i=1}^{|\mathcal{Y}|}$. Assuming identical class-conditional probabilities $p(x|y) = \hat{p}(x|y)$, between source and target domains, we derive through Bayes' theorem:

$$p(y|\mathbf{x}) \propto \frac{\hat{p}(y|\mathbf{x})p(y)}{\hat{p}(y)}. \tag{8}$$

To ensure $p(y|\mathbf{x})$ forms a valid probability distribution, we normalize it as follows:

$$p(y|x) = \frac{p(y|\mathbf{x})}{\sum_{j \in \mathcal{Y}} p(j|\mathbf{x})} = \frac{\hat{p}(y|\mathbf{x})\frac{p(y)}{\hat{p}(y)}}{\sum_{j \in \mathcal{Y}} \hat{p}(j|\mathbf{x})\frac{p(j)}{\hat{p}(j)}} \tag{9}$$

$$= \frac{\frac{\exp(v_y)}{\sum_{i \in \mathcal{Y}} \exp(v_i)} \cdot \frac{p_b^y}{\hat{p}b^y}}{\sum_{j \in \mathcal{Y}} \left[ \frac{\exp(v_j)}{\sum_{i \in \mathcal{Y}} \exp(v_i)} \cdot \frac{p_b^j}{\hat{p}_b^j} \right]} \tag{10}$$

$$= \frac{\exp(v_y) \cdot \frac{p_b^y}{\hat{p}b^y}}{\sum_{j \in \mathcal{Y}} \exp(v_j) \cdot \frac{p_b^j}{\hat{p}_b^j}} \tag{11}$$

$$= \frac{\exp\left(v_y + \log p_b^y - \log \hat{p}_b^y\right)}{\sum_{j \in \mathcal{Y}} \exp\left(v_j + \log p_b^j - \log \hat{p}_b^j\right)}. \tag{12}$$

Substituting the target class probability distribution

$$\hat{p}_b^y = \frac{1/p_b^y}{\sum_{i \in |\mathcal{Y}|} 1/p_b^i} \tag{13}$$

into Equation (12) yields the final form of $\ell_{\text{Rev}}$ in Equation (4):

$$\ell_{\text{Rev}} = -\log p(y|\mathbf{x}) = -\log \frac{\exp\left(v_y + 2\log p_b^y\right)}{\sum_{j \in \mathcal{Y}} \exp\left(v_j + 2\log p_b^j\right)}. \tag{14}$$

# B. Estimate of covariance matrices

In our imbalanced DIL framework, we employ Oracle Approximating Shrinkage (OAS) (Chen et al., 2010) to estimate class-conditional covariance matrices across sequential domains. This regularization technique addresses the small-sample-size problem in individual classes.

For class $c$ in domain $b$ with $n$ samples $\{\mathbf{x}_i\}_{i=1}^n$, the regularized covariance estimate $\hat{\Sigma}_{c,d}$ is computed as:

$$\hat{\Sigma}_{c,d} = (1 - \rho)\hat{\Sigma}_{\text{emp}} + \rho\hat{\Sigma}_{\text{prior}}, \tag{15}$$

where $\hat{\Sigma}_{\text{emp}} = \frac{1}{n-1} \sum_{i=1}^n (\mathbf{x}_i - \mu_c)(\mathbf{x}_i - \mu_c)^\top$ is the empirical covariance, $\hat{\Sigma}_{\text{prior}} = \text{tr}(\hat{\Sigma}_{\text{emp}})/d \cdot I_d$ denotes the spherical shrinkage target, and $\rho$ is the OAS shrinkage coefficient given by:

$$\rho = \frac{\left(1 - \frac{2}{d}\right) \cdot \text{tr}(\hat{\Sigma}_{\text{emp}}^2) + \text{tr}(\hat{\Sigma}_{\text{emp}})^2}{\left(n + 1 - \frac{2}{d}\right) \cdot \left(\text{tr}(\hat{\Sigma}_{\text{emp}}^2) - \frac{1}{d}\text{tr}(\hat{\Sigma}_{\text{emp}})^2\right)}. \tag{16}$$

Given the memory constraints in incremental learning, our implementation calculates OAS-regularized covariance matrices only when a class contains sufficient samples ($n \geq 10$) within a domain. The resulting matrices are aggregated through element-wise averaging to construct a shared covariance structure, effectively balancing computational efficiency with statistical reliability.

## C. More Comparative Experiments

We further conduct comparative experiments with other incremental learning methods on the Office-Home and DomainNet datasets. ROW (Kim et al., 2023) is a replay-based incremental learning method that, similar to S-iPrompt, learns a set of task-specific parameters for each task. However, it selects appropriate parameters through an out-of-distribution detection mechanism. DUCT (Zhou et al., 2024b) is a recent DIL method that addresses domain shifts by continuously merging task-specific parameters using a model merging strategy. Although these methods perform well in traditional incremental learning settings, they fail to adapt to the imbalanced DIL scenario we target. In contrast, our proposed method, DCE, achieves the best performance.

*Table 3.* Average and last performance of different methods among five task orders. The best performance is shown in bold. All methods are implemented with ViT-B/16 IN1K.

| Method | Office-Home | | DomainNet | |
|---|---|---|---|---|
| | $\bar{\mathcal{A}}$ | $\mathcal{A}_B$ | $\bar{\mathcal{A}}$ | $\mathcal{A}_B$ |
| ROW (Kim et al., 2023) | $81.1_{\pm 3.9}$ | $82.2_{\pm 0.4}$ | $58.3_{\pm 7.3}$ | $57.1_{\pm 0.3}$ |
| DUCT (Zhou et al., 2024b) | $64.7_{\pm 3.8}$ | $58.8_{\pm 5.7}$ | $26.6_{\pm 12}$ | $18.0_{\pm 6.0}$ |
| DCE | $\mathbf{84.5_{\pm 2.9}}$ | $\mathbf{84.4_{\pm 0.1}}$ | $\mathbf{64.2_{\pm 5.9}}$ | $\mathbf{63.4_{\pm 0.4}}$ |

## D. Detailed Experimental Results

In the main manuscript, we conducted experiments on benchmark datasets using five distinct task sequences and reported the average performance. To provide comprehensive insights, we present the task-specific performance for each sequence in Table 4, Table 5, Table 6, Table 7, Table 8. with full details of the task sequences documented in Appendix F.3.

*Table 4.* Average and last performance of different methods with the 1st task order in Section F.3. The best performance is shown in bold. All methods are implemented with ViT-B/16 IN1K. Methods with † indicate implementations with exemplars (10 per class).

| Method | Office-Home | | DomainNet | | CORe50 | | CDDB-Hard | |
|---|---|---|---|---|---|---|---|---|
| | $\bar{\mathcal{A}}$ | $\mathcal{A}_B$ | $\bar{\mathcal{A}}$ | $\mathcal{A}_B$ | $\bar{\mathcal{A}}$ | $\mathcal{A}_B$ | $\bar{\mathcal{A}}$ | $\mathcal{A}_B$ |
| iCaRL[†] (Rebuffi et al., 2017) | 80.2 | 84.5 | 58.2 | 58.2 | 71.8 | 75.8 | 50.7 | 51.6 |
| MEMO[†] (Zhou et al., 2023b) | 74.2 | 79.4 | 57.2 | 58.1 | 64.1 | 66.2 | 57.6 | 64.3 |
| SimpleCIL (Zhou et al., 2024a) | 71.8 | 76.2 | 38.9 | 40.6 | 62.2 | 67.2 | 62.9 | 63.3 |
| RanPAC (McDonnell et al., 2023) | 78.6 | 83.1 | 55.8 | 56.3 | 76.8 | 78.1 | 57.2 | 60.1 |
| L2P (Wang et al., 2022c) | 74.5 | 80.5 | 45.9 | 47.5 | 71.7 | 81.6 | 60.1 | 56.6 |
| DualPrompt (Wang et al., 2022b) | 72.4 | 78.7 | 44.8 | 49.4 | 68.8 | 78.4 | 65.0 | 62.0 |
| CODA-Prompt (Smith et al., 2023) | 78.6 | 83.1 | 45.8 | 46.4 | 71.5 | 82.0 | 60.8 | 65.5 |
| S-iPrompt (Wang et al., 2022a) | 78.6 | 81.1 | 56.7 | 58.1 | 63.9 | 66.3 | 59.0 | 67.7 |
| DCE | **81.7** | **84.5** | **62.2** | **63.9** | **80.8** | **84.8** | **70.2** | **72.3** |

*Table 5.* Average and last performance of different methods with the 2rd task order in Section F.3. The best performance is shown in bold. All methods are implemented with ViT-B/16 IN1K. Methods with † indicate implementations with exemplars (10 per class).

| Method | Office-Home | | DomainNet | | CORe50 | | CDDB-Hard | |
|---|---|---|---|---|---|---|---|---|
| | $\bar{\mathcal{A}}$ | $\mathcal{A}_B$ | $\bar{\mathcal{A}}$ | $\mathcal{A}_B$ | $\bar{\mathcal{A}}$ | $\mathcal{A}_B$ | $\bar{\mathcal{A}}$ | $\mathcal{A}_B$ |
| iCaRL[†] (Rebuffi et al., 2017) | 80.7 | 83.2 | 49.6 | 56.9 | 73.0 | 78.0 | 67.4 | 48.1 |
| MEMO[†] (Zhou et al., 2023b) | 78.5 | 80.2 | 46.4 | 55.3 | 66.0 | 68.4 | **84.6** | 75.0 |
| SimpleCIL (Zhou et al., 2024a) | 68.8 | 76.2 | 34.2 | 40.6 | 64.8 | 67.2 | 66.8 | 63.3 |
| RanPAC (McDonnell et al., 2023) | 78.8 | 83.7 | 50.0 | 56.4 | 78.4 | 80.0 | 66.7 | 66.1 |
| L2P (Wang et al., 2022c) | 74.8 | 81.2 | 40.1 | 42.3 | 73.1 | 81.1 | 67.9 | 61.3 |
| DualPrompt (Wang et al., 2022b) | 71.9 | 79.6 | 41.5 | 45.4 | 70.2 | 78.2 | 63.8 | 71.9 |
| CODA-Prompt (Smith et al., 2023) | 78.8 | 83.7 | 40.3 | 44.2 | 73.8 | 81.7 | 72.8 | 64.7 |
| S-iPrompt (Wang et al., 2022a) | 77.8 | 80.7 | 48.5 | 57.8 | 64.9 | 66.4 | 68.9 | 62.1 |
| DCE | **81.5** | **84.8** | **55.0** | **63.2** | **79.8** | **84.7** | 81.0 | **75.4** |

*Table 6.* Average and last performance of different methods with the 3rd task order in Section F.3. The best performance is shown in bold. All methods are implemented with ViT-B/16 IN1K. Methods with † indicate implementations with exemplars (10 per class).

| Method | Office-Home | | DomainNet | | CORe50 | | CDDB-Hard | |
|---|---|---|---|---|---|---|---|---|
| | $\bar{\mathcal{A}}$ | $\mathcal{A}_B$ | $\bar{\mathcal{A}}$ | $\mathcal{A}_B$ | $\bar{\mathcal{A}}$ | $\mathcal{A}_B$ | $\bar{\mathcal{A}}$ | $\mathcal{A}_B$ |
| iCaRL† (Rebuffi et al., 2017) | 86.2 | 83.9 | 60.1 | 54.1 | 73.2 | 77.7 | 50.0 | 51.1 |
| MEMO† (Zhou et al., 2023b) | 81.1 | 77.2 | 60.8 | 54.9 | 69.8 | 70.7 | 51.3 | 53.5 |
| SimpleCIL (Zhou et al., 2024a) | 76.7 | 76.2 | 44.5 | 40.6 | 61.4 | 67.2 | 66.3 | 63.3 |
| RanPAC (McDonnell et al., 2023) | 85.0 | 82.8 | 59.2 | 55.0 | 75.7 | 77.7 | 59.2 | 59.3 |
| L2P (Wang et al., 2022c) | 80.8 | 79.5 | 49.6 | 44.5 | 73.8 | 82.5 | 70.0 | 66.1 |
| DualPrompt (Wang et al., 2022b) | 79.2 | 78.6 | 51.0 | 49.1 | 68.6 | 77.5 | 66.1 | **68.8** |
| CODA-Prompt (Smith et al., 2023) | 85.0 | 82.8 | 47.3 | 44.5 | 72.1 | 80.1 | 69.1 | 66.4 |
| S-iPrompt (Wang et al., 2022a) | 83.9 | 80.9 | 63.0 | 57.8 | 63.2 | 66.1 | 60.3 | 60.9 |
| DCE | **86.4** | **84.5** | **66.5** | **63.1** | **79.1** | **85.3** | **70.9** | 66.4 |

*Table 7.* Average and last performance of different methods with the 4th task order in Section F.3. The best performance is shown in bold. All methods are implemented with ViT-B/16 IN1K. Methods with † indicate implementations with exemplars (10 per class).

| Method | Office-Home | | DomainNet | | CORe50 | | CDDB-Hard | |
|---|---|---|---|---|---|---|---|---|
| | $\bar{\mathcal{A}}$ | $\mathcal{A}_B$ | $\bar{\mathcal{A}}$ | $\mathcal{A}_B$ | $\bar{\mathcal{A}}$ | $\mathcal{A}_B$ | $\bar{\mathcal{A}}$ | $\mathcal{A}_B$ |
| iCaRL† (Rebuffi et al., 2017) | 86.9 | 83.3 | 60.9 | 54.6 | 72.8 | 77.1 | 69.1 | 71.9 |
| MEMO† (Zhou et al., 2023b) | 83.3 | 78.7 | 61.8 | 57.2 | 67.2 | 67.1 | **83.5** | **83.9** |
| SimpleCIL (Zhou et al., 2024a) | 81.0 | 76.2 | 51.0 | 40.6 | 63.2 | 67.2 | 69.8 | 67.1 |
| RanPAC (McDonnell et al., 2023) | 87.2 | 83.3 | 65.1 | 56.5 | 75.1 | 76.1 | 65.9 | 62.9 |
| L2P (Wang et al., 2022c) | 84.4 | 80.8 | 58.7 | 47.6 | 72.7 | 82.0 | 74.1 | 73.7 |
| DualPrompt (Wang et al., 2022b) | 84.4 | 79.9 | 60.9 | 54.6 | 79.1 | 78.2 | 69.9 | 72.4 |
| CODA-Prompt (Smith et al., 2023) | 87.2 | 83.3 | 57.2 | 46.4 | 74.3 | 82.7 | 75.2 | 71.5 |
| S-iPrompt (Wang et al., 2022a) | 85.6 | 80.5 | 65.9 | 58.1 | 62.2 | 65.7 | 72.0 | 61.6 |
| DCE | **88.5** | **84.3** | **70.3** | **63.0** | **80.7** | **84.5** | 82.4 | 76.1 |

*Table 8.* Average and last performance of different methods with the 5th task order in Section F.3. The best performance is shown in bold. All methods are implemented with ViT-B/16 IN1K. Methods with † indicate implementations with exemplars (10 per class).

| Method | Office-Home | | DomainNet | | CORe50 | | CDDB-Hard | |
|---|---|---|---|---|---|---|---|---|
| | $\bar{\mathcal{A}}$ | $\mathcal{A}_B$ | $\bar{\mathcal{A}}$ | $\mathcal{A}_B$ | $\bar{\mathcal{A}}$ | $\mathcal{A}_B$ | $\bar{\mathcal{A}}$ | $\mathcal{A}_B$ |
| iCaRL† (Rebuffi et al., 2017) | 82.0 | 82.9 | 63.4 | 55.9 | 64.4 | 71.5 | 50.0 | 49.2 |
| MEMO† (Zhou et al., 2023b) | 76.3 | 79.9 | 62.9 | 56.6 | 62.7 | 68.5 | 56.5 | 53.1 |
| SimpleCIL (Zhou et al., 2024a) | 77.7 | 76.2 | 36.7 | 40.6 | 60.8 | 67.2 | 61.9 | 63.3 |
| RanPAC (McDonnell et al., 2023) | 82.4 | 83.5 | 58.9 | 56.6 | 77.4 | 80.1 | 59.2 | 64.0 |
| L2P (Wang et al., 2022c) | 79.0 | 80.7 | 48.3 | 44.2 | 70.3 | 81.4 | 64.5 | 67.5 |
| DualPrompt (Wang et al., 2022b) | 78.2 | 78.9 | 63.4 | 55.9 | 68.6 | 76.1 | 63.5 | 59.7 |
| CODA-Prompt (Smith et al., 2023) | 82.4 | 83.5 | 47.4 | 44.1 | 72.3 | 80.4 | 61.7 | 64.8 |
| S-iPrompt (Wang et al., 2022a) | 81.4 | 80.9 | 61.2 | 57.9 | 59.2 | 64.3 | 60.8 | 64.7 |
| DCE | **84.9** | **84.4** | **67.6** | **64.2** | **80.2** | **85.0** | **68.7** | **68.9** |

# E. Ablation on the Effectiveness of Multi-Expert Design

Figure 8 presents the model performance across different class frequencies under varying expert configurations. With a single expert, $\ell_{CE}$ performs best on many-shot classes but poorly on few-shot classes; conversely, $\ell_{Rev}$ excels on few-shot classes while underperforming on many-shot ones. $\ell_{Bal}$ achieves relatively balanced results. When two experts are used, the same trend holds: $\ell_{CE} + \ell_{Bal}$ favors many-shot classes, $\ell_{Bal} + \ell_{Rev}$ benefits few-shot classes, and $\ell_{CE} + \ell_{Rev}$ provides

a more balanced trade-off. These results indicate that better class-frequency balance typically leads to improved overall performance. The three-expert design in our DCE framework can be seen as integrating the strengths of $\ell_{Bal}$ and $\ell_{CE} + \ell_{Rev}$, yielding superior results compared to single or dual expert settings.

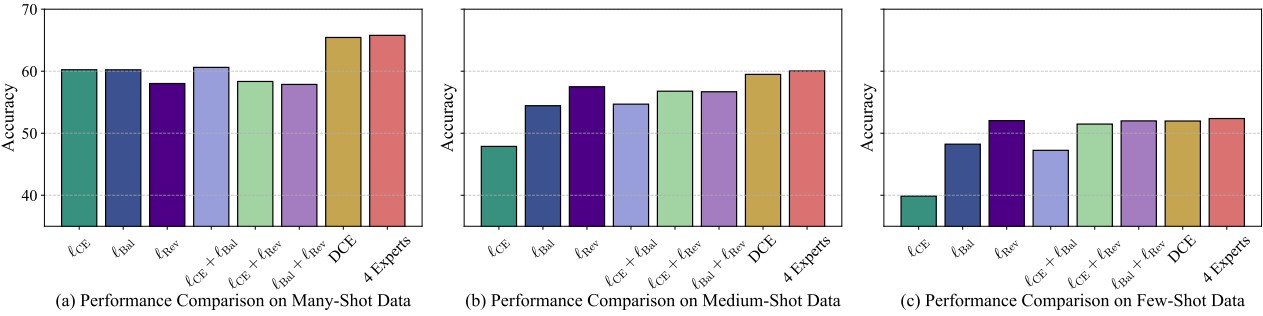

(a) Performance Comparison on Many-Shot Data   (b) Performance Comparison on Medium-Shot Data   (c) Performance Comparison on Few-Shot Data

*Figure 8.* Performance comparison under different expert configurations. Accuracy on (a) many-shot, (b) medium-shot, and (c) few-shot classes is reported on DomainNet using different expert setups.

## F. Detailed Experimental Setup

### F.1. Experimental Datasets

- **Office-Home** (Venkateswara et al., 2017) provides a controlled testbed for gradual domain shifts across four modalities: *Art* (abstract representations), *Clipart* (stylized graphics), *Product* (isolated objects), and *Real-World* (natural images). Its 15.5K images spanning 65 categories exhibit progressive distribution shifts, making it ideal for studying catastrophic forgetting under moderate-scale incremental learning scenarios.

- **DomainNet** (Peng et al., 2019) is the largest multi-domain benchmark for cross-domain continual learning, comprising 345 fine-grained object categories across six distinct visual domains: (1) *Clipart* - vector graphic illustrations, (2) *Real* - natural scene photographs, (3) *Sketch* - freehand drawings, (4) *Infograph* - information graphics with contextual elements, (5) *Painting* - artistic renderings, and (6) *Quickdraw* - time-constrained doodles. Following standard protocols (Zhou et al., 2023b), we employ the noise-filtered "Cleaned" version containing 0.6M images, with domain shifts characterized by both style and contextual variations.

- **CORe50** (Lomonaco & Maltoni, 2017) evaluates continual learning under environmental dynamics through 11 acquisition sessions (8 indoor/3 outdoor) capturing 50 household objects under varying viewpoints and illumination. Its unique RGB-D temporal sequences (300 frames/object/session) enable testing of both spatial and temporal feature stability.

- **CDDB-Hard** (Li et al., 2023) presents a continual deepfake detection challenge featuring 12 evolving forgery techniques (e.g., GAN/Neural-Texture variants) across 5 tasks. The "Hard" track (Wang et al., 2022a) introduces maximum task confusion through overlapping manipulation artifacts and progressive quality improvements, simulating real-world deception evolution. The benchmark contains 100K real/fake image pairs with temporal metadata, requiring models to maintain detection capability while adapting to emerging forgery paradigms.

### F.2. Dataset Construction.

Among the four datasets mentioned above, Office-Home and DomainNet are inherently class-imbalanced. Unlike the common practice of splitting training and testing sets proportionally, in class-imbalanced learning, an imbalanced testing set is required to evaluate the algorithms. Therefore, following the setting of Yang et al. (2022), we randomly select 10 samples from each class to construct the testing set and adopt (20, 60) and (20, 100) training samples as the demarcation for many-shot, medium-shot, and few-shot classes on Office-Home and DomainNet datasets respectively.

In contrast, the CORe50 and CDDB-Hard datasets are originally balanced, necessitating manual construction of imbalanced training sets. For CORe50, we follow the setting in Cui et al. (2019), creating different imbalanced tasks based on varying

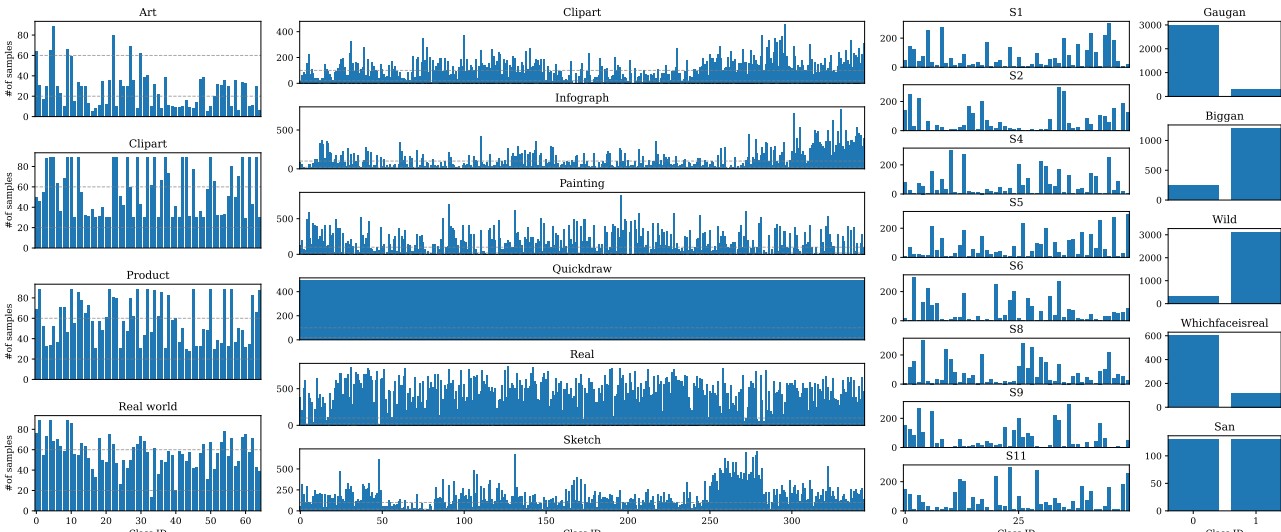

*Figure 9.* The class distribution of training samples in the four datasets. The Office-Home and DomainNet datasets are inherently imbalanced, so we constructed a class-balanced test set for evaluation. The dashed lines indicate the thresholds for the division of many-shot, medium-shot, and few-shot classes. For CORe50 and CDDB-Hard, we manually constructed imbalanced training sets.

imbalance ratios $\rho = N_{\max}/N_{\min}$, where $N_{\max}$ represents the number of samples in the majority class and $N_{\min}$ represents the number of samples in the minority class. Among the 8 training tasks, 4 are assigned an imbalance ratio of 100, and the other 4 are assigned a ratio of 50.

For the CDDB-Hard dataset, we construct class-imbalanced data by specifying the number of positive and negative samples as follows:

- *gaugan*: 3000 negatives and 300 positives

- *biggan*: 240 negatives and 1200 positives

- *wild*: 310 negatives and 3115 positives

- *whichfaceisreal*: 600 negatives and 120 positives

- *san*: 130 negatives and 130 positives

The number of training samples across all datasets is illustrated in Figure 9.

## F.3. Task Sequences

In domain-incremental learning scenarios, algorithmic performance exhibits sensitivity to domain ordering. To rigorously assess this factor, we generated five randomized permutations of domain sequences through stratified sampling, which are systematically analyzed in the main experiments. The complete domain order specifications are tabulated in Table 9, Table 10, Table 11, Table 12.

*Table 9.* Task orders of Office-Home.

| Office-Home | Task 1 | Task 2 | Task 3 | Task 4 |
|---|---|---|---|---|
| Order 1 | Art | Clipart | Product | Real_World |
| Order 2 | Clipart | Art | Real_World | Product |
| Order 3 | Product | Clipart | Art | Real_World |
| Order 4 | Real_World | Product | Clipart | Art |
| Order 5 | Art | Real_World | Product | Clipart |

*Table 10.* Task orders of DomainNet.

| DomainNet | Task 1 | Task 2 | Task 3 | Task 4 | Task 5 | Task 6 |
|---|---|---|---|---|---|---|
| Order 1 | clipart | infograph | painting | quickdraw | real | sketch |
| Order 2 | infograph | painting | sketch | clipart | quickdraw | real |
| Order 3 | painting | quickdraw | real | sketch | clipart | infograph |
| Order 4 | real | sketch | painting | infograph | quickdraw | clipart |
| Order 5 | sketch | clipart | quickdraw | real | infograph | painting |

*Table 11.* Task orders of CORe50.

| CORe50 | Task 1 | Task 2 | Task 3 | Task 4 | Task 5 | Task 6 | Task 7 | Task 8 |
|---|---|---|---|---|---|---|---|---|
| Order 1 | s11 | s4 | s2 | s9 | s1 | s6 | s5 | s8 |
| Order 2 | s2 | s9 | s1 | s6 | s5 | s8 | s11 | s4 |
| Order 3 | s4 | s1 | s9 | s2 | s5 | s6 | s8 | s11 |
| Order 4 | s1 | s9 | s2 | s5 | s6 | s8 | s11 | s4 |
| Order 5 | s9 | s2 | s5 | s6 | s8 | s11 | s4 | s1 |

*Table 12.* Task orders of CDDB-Hard.

| CDDB-Hard | Task 1 | Task 2 | Task 3 | Task 4 | Task 5 |
|---|---|---|---|---|---|
| Order 1 | wild | whichfaceisreal | san | gaugan | biggan |
| Order 2 | gaugan | biggan | wild | whichfaceisreal | san |
| Order 3 | whichfaceisreal | gaugan | wild | san | biggan |
| Order 4 | gaugan | whichfaceisreal | san | wild | biggan |
| Order 5 | wild | biggan | gaugan | san | whichfaceisreal |

## F.4. Implementation Details

Each expert is implemented as a three-layer MLP with layer widths set to $D$, $D/2$, and $|\mathcal{Y}|$, respectively. The expert selector adopts a similar architecture, except that the final layer outputs the number of experts instead of class logits. During VPT training, we set the number of prompts to 10. The batch size is fixed at 128, and we use a cosine learning rate decay schedule with an initial learning rate of 0.01. In the first training stage, we train the model for 20 epochs on the Office-Home, CORe50, and CDDB-Hard datasets, and for 30 epochs on DomainNet. In the second stage, the model is trained for 10 epochs on all datasets.

For each task, the number of stored parameters is $3 \times D \times \frac{D}{2} \times |\mathcal{Y}| + (D \times D + D \times |\mathcal{Y}|)$. The first term corresponds to the parameters of the experts, while the second term represents the stored feature statistics (i.e., the covariance matrix and the feature mean of each class). In addition, the parameter size of our expert selector is $D \times \frac{D}{2} \times 3b$, where $3b$ denotes the number of experts.

## G. Compared Methods

In this section, we introduce the methods that were compared in the main paper. **Note that we re-implement all methods using the same pre-trained model as initialization**. They are listed as follows.

- **iCaRL** (Rebuffi et al., 2017) is an exemplar-based method, which addresses forgetting by storing representative exemplars from previous tasks (i.e., in this paper, we save 10 exemplars per class) and replay them with new task data during training. It integrates knowledge distillation with exemplar retention. Beyond standard classification loss for new tasks, it enforces consistency between the current and previous models' logits via distillation loss. While this dual-objective strategy mitigates forgetting, its performance degrades under strict memory budgets due to linearly growing exemplar requirements.

- **MEMO** (Zhou et al., 2023b) is an expansion-based continual learning algorithm that adopts a dynamic architecture expansion approach, selectively growing network components to capture task-specific features while freezing existing modules. As for the implementation, we follow the original paper to decouple the network and expand the last transformer block for each new task.

- **SimpleCIL** (Zhou et al., 2024a) proposes this simple baseline in pre-trained model-based continual learning. It uses pretrained model by freezing backbone parameters and constructing a cosine classifier from class prototypes. It computes prototype vectors as feature centroids and directly assigns them as classifier weights, eliminating the need for iterative fine-tuning.

- **RanPAC** (McDonnell et al., 2023) extends SimpleCIL by randomly projecting the features into the high-dimensional space and learning the online LDA classifier for final classification.

- **L2P** (Wang et al., 2022c) is the first work introducing prompt tuning in continual learning. With the pre-trained weights frozen, it learns a prompt pool containing many prompts. During training and inference, instance-specific prompts are selected to produce the instance-specific embeddings.

- **DualPrompt** (Wang et al., 2022b) extends L2P in two aspects. Apart from the prompt pool and prompt selection mechanism, it further introduces prompts instilled at different depths and task-specific prompts. During training and inference, the instance-specific and task-specific prompts work together to adjust the embeddings.

- **CODA-Prompt** (Smith et al., 2023) aims to avoid the prompt selection cost in L2P. It treats prompts in the prompt pool as bases and utilizes the attention results to combine multiple prompts as the instance-specific prompt.

- **S-iPrompt** (Wang et al., 2022a) is specially designed for pre-trained model-based domain-incremental learning. It learns task-specific prompts for each domain and saves domain centers in the memory with K-Means. During inference, it first forwards the features to select the nearest domain center via KNN search. Afterward, the selected prompt will be appended to the input.

