# OpenReview forum: "Addressing Imbalanced Domain-Incremental Learning through Dual-Balance Collaborative Experts"
_ICML.cc/2025/Conference — ICML 2025 poster_

### Official Review · Reviewer_udEf · 2025-03-13

**Overall Recommendation:** 4

**Summary:**

The paper introduces the Dual-balance Collaborative Experts (DCE) framework to address imbalanced domain-incremental learning. The key challenges tackled are intra-domain class imbalance and cross-domain class distribution shifts. DCE employs two main components: (1) ​frequency-aware experts trained with specialized loss functions to decouple feature learning for many-shot, medium-shot, and few-shot classes, and (2) a ​dynamic expert selector that synthesizes pseudo-features via Gaussian sampling from historical class statistics to balance knowledge retention and transfer. Experiments on four benchmarks demonstrate SOTA performance, particularly in improving few-shot class accuracy while mitigating catastrophic forgetting of many-shot classes.

**Claims And Evidence:**

Yes. The primary motivation of this paper lies in its discovery that in ​DIL, ​sharing knowledge leads to improved performance for minority classes but causes forgetting in majority classes. Conversely, ​not sharing knowledge reduces forgetting in majority classes but fails to enhance the performance of minority classes. This motivation is clear, intuitively correct, and empirically validated through the experimental results presented in ​Figure 2.

**Essential References Not Discussed:**

Currently, no papers related to this field that have not been cited have been identified.

**Experimental Designs Or Analyses:**

The paper conducts comprehensive experiments on common benchmark datasets and compares the proposed method with SOTA DIL approaches, including ​shared prompt and ​domain-specific prompt methods, as well as ​exemplar-based methods. The experiments are evaluated using multiple metrics, and the results demonstrate superior performance, highlighting the robustness of the proposed method.

**Methods And Evaluation Criteria:**

Yes. The dual-phase training is logically sound. Frequency-aware losses and Gaussian-sampled pseudo-features are appropriate for addressing class imbalance and distribution shifts. Benchmarks are well-chosen, and metrics align with the problem’s requirements.

**Other Comments Or Suggestions:**

Refer to weakness and questions.

**Other Strengths And Weaknesses:**

Strengths:
1. One significant contribution of this work lies in proposing a new ​imbalanced DIL setting, where class distributions within each domain are both imbalanced and distinct across domains. This is a highly practical issue that has been largely overlooked in prior research.
2. The paper is ​well-motivated and its greatest strength is the empirical dissection of existing methods’ failures in imbalanced DIL. Specifically, it reveals the inherent ​trade-off between leveraging shared knowledge to improve few-shot class performance and preserving historical knowledge to mitigate catastrophic forgetting of many-shot classes. This finding is highly insightful and contributes significantly to the field.
3. The proposed DCE aligns seamlessly with the authors' motivation. The ​frequency-aware experts and ​dynamic expert selector are ​organically integrated to address the two key challenges in imbalanced DIL. The method proposed in this paper is ​not complex, yet it is ​highly intuitive, ​innovative, and ​feasible in practice.
4. The experiments in this paper are ​thorough and comprehensive. The authors compare the proposed method with multiple SOTA approaches across ​several benchmarks and provide a detailed analysis of the results. These experiments effectively demonstrate the ​superior performance and ​effectiveness of the proposed method.
5. The structure of the paper is clear, making it easy for readers to follow and understand.

Weaknesses：
1. Some of the experimental settings in the paper require ​further clarification. The criteria for partitioning classes into many-shot, medium-shot, and few-shot groups are not explicitly defined.  And the design of the expert networks need further elaboration.
2. Figure 2 requires further explanation to improve its clarity and ease of understanding. Specifically, the rationale for training the prompt parameters $\theta_1$ ​only during the first task via VPT is not explicitly discussed. It is recommended to provide a ​more detailed explanation of the figure in the ​caption of Figure 2.
3. The paper introduces ​Class Performance Drift (CPD), inspired by the ​Forgetting Measure, to evaluate the change in class performance during training. However, the description of how CPD is calculated appears somewhat unclear. Providing a ​mathematical formula to define CPD would significantly enhance the readability and clarity of the paper.
4. The ​source code is not included in the supplementary material. Publicly releasing the code would enhance reproducibility and adoption.

**Questions For Authors:**

1. How does DCE ensure ​diversity among experts trained for each domain?
2. The paper explicitly defines many-shot, medium-shot, and few-shot classes for Office-Home and DomainNet but does not apply this categorization to CORe50 and CDDB-Hard. Why is this distinction omitted for these datasets?
3. In ​Figure 5, why does the accuracy on ​DomainNet decrease as the training progresses, while the accuracy on the ​CORe50 dataset generally increases?
4. In ​Figure 6, according to the ​Class Performance Drift metric, the CPD of ​SimpleCIL appears to be lower than that of the proposed ​DCE method, yet its overall performance is inferior to other methods. How can this phenomenon be explained?
5. In the experimental section, are there additional results or analyses that could ​further support the paper’s motivation?

**Relation To Broader Scientific Literature:**

Compared to existing methods, this paper places greater emphasis on addressing the imbalance issues inherent in DIL. It provides a detailed analysis of the limitations of prior DIL approaches in the context of imbalanced DIL scenarios and offers a targeted solution to these challenges.

**Theoretical Claims:**

Yes. Equation (3) derivation is correct under the assumption that the target distribution $p^​(y)$ is inversely proportional to the source distribution $p(y)$.
OAS covariance regularization is a valid approach for stabilizing imbalanced class statistics.

---

> ### Author Rebuttal · Authors · 2025-04-01
>
> **[Reviewer xasy (Claims1-3), N4e7 (Q4), udEf (W1, Q2)]: Questions on many/medium/few-shot class division**
>
> Thank you for your questions. Reviewer xasy and N4e7 raised similar concerns, which might be due to our explanation of dataset division being placed in Appendix E.2. The division of many/medium/few-shot classes is solely for the convenience of description and experimental results presentation in the paper and is not directly related to our proposed algorithm. The loss functions of the three frequency-aware experts rely only on the class frequencies in the training data and do not depend on the class division. In our experiments, following the settings in [1], we used (20, 60) and (20, 100) as thresholds to divide the many/medium/few-shot classes within each domain of the Office-Home and DomainNet datasets. And we sample class-balanced test sets from the corresponding domains. Unlike these two datasets, CORe50's test set is not sampled from each task’s data but consists of three outdoor sessions, making it impossible to directly align test set classes with the corresponding domain class divisions. Additionally, each task in CDDB-Hard is a binary classification problem. As a result, these two datasets do not undergo many/medium/few-shot class division. We will provide a more detailed explanation in the revised version to prevent misunderstandings.
>
> **Q1**: The three frequency-aware experts in DCE are trained with different loss functions, each favoring classes of different frequencies. The distinct optimization objectives lead to diversity in predictions among the three experts. For further analysis, please refer to our response to Reviewer xasy under "Effectiveness of multiple experts."
>
> **Q3**:  As mentioned above, CORe50's test set is derived from three outdoor sessions. Thus, forgetting has a relatively minor impact on this dataset, whereas knowledge sharing across tasks is more crucial. Most methods exhibit performance improvements as training progresses. However, prompt-specific methods, such as S-iPrompt, lack knowledge sharing across tasks, resulting in inferior performance compared to other approaches. The results on CORe50 further demonstrate that our method enables better knowledge sharing compared to prompt-specific methods like S-iPrompt.
>
> **Q4, W3**:  Class Performance Drift (CPD) is a new metric we introduced, inspired by the forgetting measure in CIL. CPD quantifies the performance change of test data from domain $b$ after training on domain $b$ is completed compared to the final model trained on all domains. It is computed as $\text{CPD} = \frac{1}{B-1}\sum_{b=1}^{B-1}(\mathcal{A}_b-\mathcal{A}_B)$. SimpleCIL exhibits a lower CPD because it lacks a training process and only constructs class prototypes for classification in each task. It sacrifices model performance in exchange for lower CPD.
>
> **Q5**：Our motivation can be further explained through the CPD results. As shown in Figure 6, shared prompt-based methods exhibit more significant performance degradation in many-shot and medium-shot classes but greater performance improvements in few-shot classes. In contrast, domain-specific prompt-based methods show the opposite trend. Our DCE method strikes a balance between these two extremes, aligning well with our intended motivation.
>
> **W2**: For additional clarification on Figure 2, please refer to our response to Reviewer N4e7 (W3, Q1). On one hand, we apply VPT to adapt the model to each task. On the other hand, we need to sample in the feature space to train the expert selector. Therefore, we choose the MLP trained after the feature encoder as the expert for each task and fix the feature encoder in subsequent tasks to maintain the stability of the feature space used for sampling. Some comparison methods, such as RanPAC, also adopt a similar approach, where VPT fine-tuning is only conducted for the first task.
>
> **W4**: We assure that the complete code will be released after the paper is accepted.
>
> [1] On Multi-Domain Long-Tailed Recognition, Imbalanced Domain Generalization and Beyond.

---

> > ### Comment · Reviewer_udEf · 2025-04-07
> >
> > Thank you for addressing the concerns. I will raise my score accoridngly.

---

### Official Review · Reviewer_eXBQ · 2025-03-13

**Overall Recommendation:** 3

**Summary:**

This paper introduces a practical task, Imbalanced Domain-Incremental Learning, which involves both intra-domain class imbalance and cross-domain class distribution shifts. To address this task, the authors propose the Dual-Balance Collaborative Experts (DCE) framework, which leverages a multi-expert collaborative approach. The method demonstrates superior performance in a series of extensive experiments.

**Claims And Evidence:**

Yes, the method proposed in this paper solves the introduced task.

**Essential References Not Discussed:**

None

**Experimental Designs Or Analyses:**

The authors have conducted comprehensive experiments, effectively demonstrating the superiority of the proposed method compared to existing CIL/DIL approaches. However, I believe there are still some shortcomings in the experiments:

1. The proposed method appears to be more complex than existing baselines, and while the authors claim that it "significantly reduces computational overhead," a comparison with existing methods in terms of time and space costs would be helpful for clarification.

2. While the authors address the issue of class imbalance, the baselines used for comparison are all conventional CIL/DIL methods, which might not provide a fair comparison. This does not fully highlight the ability of the proposed method to handle class imbalance. It would be beneficial to include a wider range of baselines or more robust methods in the comparison.

3. The experiments focus solely on accuracy (Acc) comparisons and exploration. It would be valuable to conduct more in-depth experiments using a broader set of metrics to gain additional insights into the method. For example, exploring the practicality of the method with three different experts could provide more comprehensive results.

**Methods And Evaluation Criteria:**

Yes, the proposed method has real-world application significance.

**Other Comments Or Suggestions:**

None

**Other Strengths And Weaknesses:**

1. The paper is well-motivated and seems to be reproducible.
2. The process of the method is clear and I think it is reproducible.

However, I believe the authors need to further clarify the innovation of their method. The proposed approach lacks a clear sense of novelty and seems more like a combination of "better solutions," especially with the introduction of the Frequency-Aware Experts. A targeted comparison and explanation, along with task-specific reasoning, might help highlight the true innovation of the method.

**Questions For Authors:**

The authors' summary of the two challenges in the task seems somewhat overlapping. In my understanding, intra-domain class imbalance and cross-domain class distribution shifts can be viewed as aspects of the broader issue of class imbalance. The second challenge appears to address catastrophic forgetting. This is just my personal interpretation, and I have some considerations regarding the method's innovation and the completeness of the experiments. While I currently hold a positive evaluation, my opinion might be swayed by the assessments of other experts in continual learning and imbalanced learning.

**Relation To Broader Scientific Literature:**

The authors have extended the existing DIL task based on the setting of class imbalance, which demonstrates broad relevance to the existing literature.

**Theoretical Claims:**

There is no much mathematical theory in the paper.

---

> ### Author Rebuttal · Authors · 2025-04-01
>
> Thanks for your suggestions.
>
> **[Reviewer xasy(Q), N4e7(E1)] Scalability**
>
> We analyze scalability from two aspects: memory consumption and computational efficiency.
> - Memory Consumption:
>   In incremental learning, it is common to retain certain past task information to mitigate forgetting. Among the compared methods, L2P maintains a prompt pool and corresponding keys per task. Our DCE  retains three expert networks, class mean/covariance, and a shared expert selection network.
>   To provide a quantitative comparison, we report the number of parameters retained outside the feature encoder for different methods at the end of the last task on the Office-Home dataset, as shown in Table 2 of the anonymous link [1].
>
> - Computational Efficiency:
>   As shown in Table 2, DCE requires more parameters to be learned and stored compared to some baseline methods, primarily due to the presence of multiple experts in our model. However, our approach remains computationally efficient due to the following reasons:
>   - In contrast to baseline methods such as L2P, DualPrompt, which require **two forward passes through the feature encoder** during both training and inference, DCE requires only **a single forward pass**. This is because baseline methods first obtain an embedding from the original feature encoder, use this embedding to select prompts, and then recompute a second forward pass with the selected prompts. In contrast, DCE directly processes the input in a single pass, significantly reducing training and inference costs.
>   - After the first task, only expert parameters are updated, and gradients do not propagate through the feature encoder, avoiding the need to construct a computational graph over it. Thus DCE reduces the computational burden during training compared to other methods.
>   - To further validate our efficiency, we conducted experiments on an RTX 3090 GPU and recorded the average time per batch during training and inference for different methods under the same batch size. The results are reported in Table 2 of the anonymous link [1].
>
> In the revised version, we will supplement our analysis of scalability, memory consumption, and computational efficiency to provide a more comprehensive discussion.
>
>
> **E2**:  Since our work is the first to explore the PTM-based imbalanced DIL problem, we primarily compare against commonly used PTM-based CIL/DIL approaches. To ensure fairness, we also modified the baseline methods by replacing their cross-entropy loss with balanced cross-entropy loss, a widely adopted approach for class imbalance. The results, shown in the third column of Figure 5, demonstrate that our method still outperforms the baselines. Additionally, we compared our approach with DUCT [2], a recently published DIL method in CVPR 2025, and reported the results in Table 3 of the anonymous link [1]. Our method also achieves superior performance over this latest approach.
>
> **E3**：Besides accuracy, we also report other evaluation metrics. For instance, inspired by the forgetting measure in CIL, we propose a new metric, CPD (shown in Figure 6), to track the performance variation of each class throughout training. For further analysis and discussion on different experts, please refer to our response to Reviewer xasy in the "Effectiveness of multiple experts section".
>
> **Weakness and Questions**：
> Regarding novelty, we address an overlooked yet prevalent issue in DIL: intra-domain class imbalance and cross-domain class distribution shift, naturally present in Office-Home and DomainNet. We analyze why existing incremental learning paradigms struggle with these challenges—shared prompt paradigms suffer from catastrophic forgetting in many-shot classes while benefiting few-shot ones, whereas Domain-specific prompts mitigate forgetting but fail to help few-shot classes. Our key insight, previously unexplored, drives our approach: balancing knowledge retention and new task learning in imbalanced DIL. If you have further concerns, please refer to our response to Reviewer N4e7 [W2, Q2]. Instead of simply combining "better solutions," our method integrates the strengths of both paradigms. Reviewer xasy considers our work a "novel framework," and Reviewer udEf also acknowledges our contributions.
>
> As you pointed out, intra-domain class imbalance and cross-domain class distribution shift are not independent but rather different perspectives on the same phenomenon. Correspondingly, the two components of our DCE framework—frequency-aware experts and dynamic expert selector—are not separate but are designed to work together to tackle these challenges. This further reinforces that our method is not merely a combination of "better solutions."
>
> [1]https://docs.google.com/spreadsheets/d/1lTmW7KBOpFPDM7FInYMTwlwP-ULQb_13-r8Vl7EfT3M/edit?usp=sharing This link contains all tables referenced in the rebuttal.
>
> [2] Dual Consolidation for Pre-Trained Model-Based Domain-Incremental Learning. CVPR2025.

---

### Official Review · Reviewer_N4e7 · 2025-03-14

**Overall Recommendation:** 2

**Summary:**

This paper addressed the problem of imbalanced domain-incremental learning, where the imbalance includes intra-domain class imbalance and cross-domain class distribution shifts. A Dual-Balance Collaborative Experts (DCE) framework is proposed, which first trains frequency-aware expert networks separately to mitigate intra-domain imbalance, and then employs a dynamic expert selector with the synthesized balanced pseudo-features to balance knowledge retention and transfer. Experiments were conducted on four benchmarks, including DomainNet, CDDB-Hard.

**Claims And Evidence:**

partially, see below of weaknesses

**Essential References Not Discussed:**

no

**Experimental Designs Or Analyses:**

yes

**Methods And Evaluation Criteria:**

yes

**Other Comments Or Suggestions:**

none

**Other Strengths And Weaknesses:**

Strengths:
1. The paper is well organized and easy to read.
2. The proposed method is evaluated on four benchmarks, and demonstrates better results than the compared methods.

Weaknesses:
1. The two issues addressed in this paper are two common ones for domain-incremental learning, however, it is unclear about the innovations of the proposed method in dealing with them compared to existing works. It also lacks in-depth analysis of why the proposed solution could achieve better results when dealing with these problems.
2. The paper did not describe clearly why it can preserve knowledge of many-shot classes while integrating few-shot patterns for new domains, why integrating new patterns for the new domains will not influence the learned patterns in the previous domains.
3. While existing works may not explicitly claim about dealing with class imbalance, this issue is implicitly considered. Claiming this work as the first one to explore this issue is exaggerated. In addition, there is no solid ablation to support this argument. The good property of DCE in Figure2 may be due to the better baseline in a single-domain setting, i.e., better results in b1 (no domain-incremental is involved) than its competitors.


==================== post rebuttal =======================

After read the rebuttal and other reviews, the reviewer maintains the initial recommendation.

**Questions For Authors:**

About Fig2, does the specific classes falling into {many-shot, medium-shot, few-shot} differs across the domains or not? If not, then it did not reflect the claimed class imbalance issue encountered in the DIL setting. If it is, it needs to clarify and is better to provide an analysis about how the dynamics of these classes influence the final results. For instance, the ratio of few-shot classes in the whole dataset, or the specific classes in a given ratio of few-shot classes.

It would be better if the authors can provide some illustrations to help understanding how the proposed method works in dealing with mentioned two issues.

The Equ.(4) is not correct, as the three experts were trained independently using one of three terms in Equ.(4). In other words, the network was never trained using Equ.(4).

The appendix described how to construct the imbalanced training sets for CORe50 and CDDB-Hard, it is still unclear how to divide them into many/medium/few-shot classes.

**Relation To Broader Scientific Literature:**

The paper could have broader scientific impact on class-imbalanced learning, domain-incremental learning, multimodal continual learning, and mixture of experts.

**Theoretical Claims:**

yes

---

> ### Author Rebuttal · Authors · 2025-04-01
>
> Thank you for your suggestions to help us improve the paper.
>
> **W2**: Our paper does not claim that "integrating new patterns for the new domains will not influence the learned patterns in previous domains." Instead, our goal in imbalanced DIL is to strike a balance between the two.
>
> As detailed in Sec 3.2, existing PTM-based methods follow two main paradigms:
>   - Shared prompt: Select prompts from the prompt pool based on sample features and use them with the feature encoder for prediction. This allows old tasks to use new prompts, risking forgetting in many-shot classes but potentially improving few-shot classes.
>
>   - Domain-specific prompts: Each task has a dedicated prompt, applied based on the test sample’s domain.  While this prevents forgetting when domain prediction is accurate, few-shot classes cannot benefit from cross-domain knowledge sharing.
>
> This limitation is illustrated in Fig 2. Our method tackles these challenges by balancing old knowledge retention and new knowledge learning. DCE trains multiple frequency-aware experts per task using different loss functions. This approach not only mitigates intra-class imbalance but also, like domain-specific prompts, preserves full expert parameters for each task, preventing forgetting caused by uncontrolled parameter combinations in shared prompt methods.
>
> To enhance cross-task knowledge sharing, we introduce an expert selector in a two-stage training process. It assigns expert combinations to class samples within each domain, mitigating few-shot class performance degradation due to limited cross-domain knowledge sharing in domain-specific prompts. If an old domain has learned strong patterns, it prioritizes its expert; otherwise, if the new expert performs better, the expert selector adjusts the weight accordingly. The expert selector balance old and new knowledge because it is trained by sampling an equal number of features across tasks and classes. This ensures appropriate expert weight allocation across domains for effective integration. Since the feature encoder remains fixed, the sampled features stay stable during expert selector training, unaffected by domain shifts.
>
> **W1**: Intra-domain class imbalance and cross-domain class distribution shifts are two common challenges in DIL. For example, the datasets used in our experiments, Office-Home and DomainNet, naturally exhibit such distributions. However, existing works often overlook this issue and typically split test data based on the class distribution of the training set. In our setting, we construct a class-balanced test set to ensure a fair evaluation across different classes.
>
> Additionally, as detailed in our response to W2, we explicitly discuss how our method differs from existing approaches and why it achieves better performance.
>
> **W3**: The existing works [1][2][3]  focus on the exemplar-based scenario, treating stored samples as imbalanced classes, or on the CIL. Our paper primarily addresses the exemplar-free DIL setting, which differs from these works. We will refine our claim for greater precision.
>
> Regarding Figure 2, the stronger baseline on $b_1$ demonstrate our method’s effectiveness in handling intra-domain class imbalance. To assess the impact of cross-domain class distribution shifts on different methods, it is essential to analyze the performance trends of various classes throughout the incremental training process.
>
> **Q1**: Figure 2 illustrates the performance trajectory of test samples from the first domain throughout training. The categorization into many-shot, medium-shot, or few-shot varies across domains. In Figure 4, we present the number of samples per class for each task, ensuring class IDs are consistent across subfigures. Additionally, Fig 6 analyzes class performance drift, capturing performance trends across domains for different test groups. In the revised version, we will refine this explanation and enhance Fig 4 by adding threshold markers on the vertical axis and specifying the number of categories in each group for each domain.
>
> **Q2**: The new illustration is in the second page of link [4].
>
> **Q3**: The concern regarding the correctness of Eq.(4) appears to stem from a misunderstanding. We formulate Eq.(4) as a unified loss because, during training, all three loss functions are computed on the same batch of input $x$ rather than being optimized independently.
>
> **Q4**: Please refer to our response to Reviewer udEf in the section "Questions on many/medium/few-shot class division."
>
> **We will adjust the paper according to your suggestions to make it clearer and more readable.**
>
> [1] Rethinking Class-Incremental Learning from a Dynamic Imbalanced Learning Perspective
>
> [2] Long-Tailed Class Incremental Learning
>
> [3] Long-Tail Class Incremental Learning via Independent SUb-Prototype Construction
>
> [4] https://docs.google.com/spreadsheets/d/1lTmW7KBOpFPDM7FInYMTwlwP-ULQb_13-r8Vl7EfT3M/edit?usp=sharing

---

### Official Review · Reviewer_xasy · 2025-03-16

**Overall Recommendation:** 2

**Summary:**

The paper introduces Dual-Balance Collaborative Experts (DCE), a novel framework designed to address two key challenges in domain-incremental learning (DIL) under class-imbalanced conditions:

1. Intra-domain class imbalance, where some classes have significantly fewer samples than others within the same domain, leading to underfitting in few-shot classes.

2. Cross-domain class distribution shifts, where the distribution of classes changes across domains, making it difficult to balance knowledge retention and adaptation.

The key techniques proposed are frequency-aware expert modules, which handle different class frequency levels (many-shot, medium-shot, and few-shot), and a dynamic expert selector, which addresses cross-domain shifts. Experiments on ViT pre-trained on ImageNet-21K and ImageNet-1K demonstrate DCE's effectiveness over the baselines.

**Claims And Evidence:**

1. The DCE framework mitigates intra-domain class imbalance and improves few-shot learning performance:
     1) It is unclear whether one of the three loss functions contributes more to the improvement or if the improvement results from their combination.
     2) It is also unclear whether the three different loss frequencies are sufficient.
     3) The definitions of many-shot, medium-shot, and few-shot classes are not clearly explained.

**Essential References Not Discussed:**

Refer to Relation to Broader Literature

**Experimental Designs Or Analyses:**

The experiment designs make sense

**Methods And Evaluation Criteria:**

The method and evaluation criteria make sense for the problem

**Other Comments Or Suggestions:**

Minor typo: in line 267, "capabilities" appeared twice.

"Specifically, due to cross-domain label distribution shift, the experts trained in the new domain may have stronger representation **capabilities** **capabilities** for few-shot classes in the old domain."

**Other Strengths And Weaknesses:**

NA

**Questions For Authors:**

Each task introduces 3 modules. Therefore, it's not scalable if there are many tasks. Please discuss scalability, memory consumption, and computational efficiency.

**Relation To Broader Scientific Literature:**

Contribution to CL:

- The proposed technique appears to be applicable to other CL problems, such as class-incremental learning. Therefore, the paper should also compare this method against existing continual learning approaches, such as [1].
- [1] examines the effect of out-of-distribution (OOD) detection in continual learning and demonstrates the learnability of CL. OOD detection can be useful for expert selection, potentially enhancing the learnability of DIL.

Limitation:
- Since the paper merely proposes a technique without theoretical justification, its impact is limited.

[1] Learnability and Algorithm for Continual Learning. ICML 2023

**Theoretical Claims:**

There is no theoretical justification/study in this paper except the derivation of a loss function (Eq. 3). I checked the derivation (Appendix A) and did not see any issue.

---

> ### Author Rebuttal · Authors · 2025-04-01
>
> We are grateful to the reviewer for their suggestions, which enhanced our paper.
>
> **[Reviewer xasy(Claims1-1,Claims1-2), eXBQ(E3)] Effectiveness of multiple experts.**
>
> In Section 5.3, we discussed the effectiveness of multiple experts. To address the reviewers' concerns more thoroughly, we conducted additional ablation studies, **with the results presented in Table 1 of the anonymous link [1]**.
>
> As shown in results, when using only a single expert, the expert trained with $l_{bal}$ performs best. When employing two experts, the combination of $l_{ce} + l_{rev}$ yields the best results. The experimental results indicate that more balanced experts tend to achieve better performance. Specifically, experts trained with $l_{ce}$ and $l_{ce} + l_{bal}$ are more beneficial for many-shot classes, whereas those trained with $l_{rev}$ and $l_{rev} + l_{bal}$ perform better for few-shot classes. Our three-expert approach can be viewed as a combination of $l_{bal}$ and $l_{ce} + l_{rev}$, which achieves superior results compared to using a single expert or two experts.
>
> To further investigate whether three losses are sufficient, we introduced a fourth expert with $l_4=-\log \frac{\exp \left(v_{y}^{4}+3 \log p_{b}^{y}\right)}{\sum_{j \in|\mathcal{Y}|} \exp \left(v_{j}^{4}+3 \log p_{b}^{j}\right)}$. The results indicate that the fourth expert provides minimal improvement and may even have adverse effects. Therefore, we conclude that three experts are a more suitable choice. We will incorporate this analysis into the revised version of our paper.
>
>
> **Claims1-3**: Please refer to our response to Reviewer udEf in the section "Questions on many/medium/few-shot class division."
>
> **Relation 1**：Thank you for your insightful suggestion.
> - Our method is not well suited for direct comparison with ROW under a CIL setting.
>   - Motivation-wise, ROW is similar to methods we analyzed, such as S-iprompt. Row train task-specific parameters and use an OOD detector to select them during inference. This approach works well in CIL, where tasks have different label spaces, so using the corresponding task’s parameters usually leads to the best performance. However, our method is designed for imbalanced DIL. In DIL, tasks share the same label space. And due to class imbalance, few-shot classes in one task can benefit from data from other tasks. This means using task-specific parameters is may not optimal for few-shot classes. Selecting task-specific parameters for prediction mitigates many-shot class forgetting in past tasks but fails to enhance few-shot class performance using knowledge from new tasks. One of our motivation is to address this issue, which conflicts with the CIL.
>
>   - ROW is a replay-based CIL method, whereas our approach try to solve a replay-free DIL problem. Since we do not store original samples, our method is difficult to compare with ROW under the CIL setting.
>
> - Following your suggestion, **we conducted a comparison with ROW under the imbalanced DIL setting, and the results are reported in the Table3[1].** The original ROW does not outperform our approach. **We will include ROW as a baseline method in the revised version.**
> - In future research, combining our approach with ROW could be a promising direction for addressing imbalanced DIL.  For example, modifying ROW’s OOD detection mechanism so that if past task samples can still be well classified under new parameters, they are not treated as OOD samples. However, this would require further exploration in future work.
>
>
> **Relation 2**：The contributions of our work are twofold. First, we identify two commonly overlooked challenges in DIL: intra-domain class imbalance and cross-domain class distribution shifts. These challenges naturally arise in commonly used DIL datasets such as DomainNet and OfficeHome, yet prior research has largely ignored them. We also highlight a crucial difference between two prevalent incremental learning paradigms in handling this issue (discussed in Section 3.2): shared prompt methods facilitate knowledge sharing across tasks, leading to forgetting for many-shot classes while improving few-shot class performance, whereas domain-specific prompt methods exhibit the opposite effect. Second, based on our findings, we propose a novel solution DCE with two coupled components, integrating the strengths of both paradigms to address the two challenges. Given that incremental learning and class imbalance learning are both fundamental topics in machine learning, we believe our discoveries contribute meaningfully to the ML community. Reviewer udEf also acknowledged the significance of our contributions.
>
> **Suggestions**: We appreciate your suggestions and will address them in the revised version.
>
> **Questions**: Please refer to our response to Reviewer eXBQ under the Scalability section.
>
>
> [1]https://docs.google.com/spreadsheets/d/1lTmW7KBOpFPDM7FInYMTwlwP-ULQb_13-r8Vl7EfT3M/edit?usp=sharing This link contains all tables referenced in the rebuttal.

---

### Decision · Program_Chairs · 2025-05-01

**Decision:**

Accept (poster)

**Comment:**

The paper introduces Dual-Balance Collaborative Experts (DCE), a novel framework designed to address the challenges of domain-incremental learning (DIL) under class-imbalanced conditions. These challenges include intra-domain class imbalance, where some classes have significantly fewer samples within the same domain, and cross-domain class distribution shifts, where the class distribution changes across domains. DCE employs frequency-aware expert modules to handle different class frequency levels (many-shot, medium-shot, and few-shot) and a dynamic expert selector to balance knowledge retention and adaptation across domains. The method is evaluated on ViT pre-trained on ImageNet-21K and ImageNet-1K, demonstrating its effectiveness over baselines.

Some of the main strengths of the paper that are highlighted by the reviewers are:
1. The paper tackles two key challenges in domain-incremental learning (DIL) under class-imbalanced conditions: intra-domain class imbalance and cross-domain class distribution shifts.
2. The paper introduces frequency-aware expert modules, which handle different class frequency levels (many-shot, medium-shot, and few-shot).
3. The paper introduces a dynamic expert selector, which addresses cross-domain shifts.
4. One of the main strengths of this paper is the empirical results on existing methods’ failures in imbalanced DIL. Specifically, it reveals the inherent trade-off between leveraging shared knowledge to improve few-shot class performance and preserving historical knowledge to mitigate catastrophic forgetting of many-shot classes.

The reviewers highlighted several weaknesses:
1. There was a lack of clarity when it came to the different class divisions, namely, what constituted many, medium, and few shot.  This was addressed in the author rebuttal.
2. There was insufficient discussion on scalability given that there are multiple expert modules.  This was addressed sufficiently in the author rebuttals.
3. The reviewers pointed out that there were insufficient baselines, and the authors included two new baselines in the rebuttal.
4. Given that it seems that the main issues that were raised by the reviewers were addressed in the rebuttal, I think this paper merits acceptance.